# Can We Use CAR-T Cells to Overcome Immunosuppression in Solid Tumours?

**DOI:** 10.3390/biology14081035

**Published:** 2025-08-12

**Authors:** Julia Gwadera, Maksymilian Grajewski, Hanna Chowaniec, Kasper Gucia, Jagoda Michoń, Zofia Mikulicz, Małgorzata Knast, Patrycja Pujanek, Amelia Tołkacz, Aleksander Murawa, Paula Dobosz

**Affiliations:** 1Faculty of Medicine, Poznan University of Medical Sciences, 61-701 Poznan, Poland; j.gwadera.era@gmail.com (J.G.); kaspergucia@gmail.com (K.G.); zofiamikulicz2@gmail.com (Z.M.); malgosia.knast@gmail.com (M.K.);; 2Department of Patomorphology, Poznan University of Medical Sciences, 61-701 Poznan, Poland

**Keywords:** CAR-T-cell therapy, CAR-T cells, solid tumours, tumour microenvironment, immunosuppression

## Abstract

Chimeric antigen receptor (CAR)-T-cell therapy has shown high efficacy in hematologic malignancies but remains limited in solid tumours due to the immunosuppressive tumour microenvironment and physical barriers. This review discusses emerging engineering and combinatorial strategies aimed at enhancing CAR-T-cell function, persistence, and infiltration in solid tumours, as well as possible future direction and scientific strategies aimed at overcoming existing barriers.

## 1. Introduction

Chimeric antigen receptor (CAR)-T-cell therapy is a relatively new strategy that uses modified T lymphocytes. CAR-T-cell therapy is classified as an advanced therapy medicinal product (ATMP) regulated under European Medicines Agency (EMA) and Food and Drug Administration (FDA) law. ATMPs belong to a new era of personalised medicine, in which the treatment is tailored to the individual needs of a patient. The term *chimeric* refers to the combination of composing parts of the receptor from its different sources. CARs are genetically engineered molecules that provide immune effector cells with specific targeting abilities, enabling them to recognise and bind to tumour cells with the precision of a monoclonal antibody [1]. Their aim is to redirect the immune system to identify and eliminate cancer cells to achieve cancer remission. CAR-T cells can be derived from either the patient (autologous) or a healthy donor (allogeneic). In both cases, the starting material consists of donated lymphocytes, which are the initial cell population used for engineering T lymphocytes with chimeric antigen receptors to create the therapeutic CAR-T cells. Autologous CAR-T cells are collected from the patient’s blood through leukapheresis, while allogeneic CAR-T cells are obtained from donor blood. These engineered receptors, expressed on various T-cell subsets, enable specific antigen recognition independent of the major histocompatibility complex (MHC). In theory, CAR-T cells can be designed to target almost any tumour-associated antigen (TAA) and even non-tumour antigens such as those found in fungal pathogens [2]. Given their recent increasing usage and global interest in cancer treatment, in this review, we decided to consider CAR-T-cell therapy as a potential weapon supporting immunosuppression breakdown in solid tumours.

### CAR-T-Cell Therapy—Overview and Historical Background

The (CAR)-T structure is relatively simple; however, recent modifications can further expand this basic model, depending on the purpose of the treatment. The basic form consists of an ectodomain, a transmembrane domain, and an endodomain [3,4]. The ectodomain is responsible for antigen binding. It is made from a single-chain variable fragment (ScFv) composed of a light and a heavy variable monoclonal antibody fragment connected by a hinge region to a transmembrane domain. This gives CAR-T cells the ability to recognise specific antigens, such as cluster of differentiation (CD)19, B-cell maturation antigen (BCMA), CD20, and CD30, without requiring HLA presentation. Longer hinges provide more flexibility and better access to membrane-proximal antigens. Shorter hinges offer less flexibility and target antigens further away [3]. The transmembrane domain connects the ectodomain to the endodomain. It plays a key role in stabilising CAR expression and enabling signalling but has not been studied as much as other CAR parts. To overcome this, researchers have developed synthetic transmembrane domains, known as programmable membrane proteins (proMPs) [3,4]. These are custom-designed sequences that form stable membrane structures, allowing precise control over CAR activity. CARs built with proMPs (called proCARs) show more predictable function and reduced inflammatory cytokine release, without affecting binding or signalling, making them a promising option for safer, more effective CAR T therapies [3]. The endodomain is also referred to as the intracellular signalling domain or cytoplasmic tail. It is built to mimic the natural TCR and contains TCR (CD3ζ—a key transmitter of signals from the TCR) co-receptor, which includes three ITAMs (immunoreceptor tyrosine-based activation motifs) that convey primary signals. In addition to CD3ζ, the endodomain also includes co-stimulatory molecules, such as CD28, tumour necrosis factor receptor (TNFR) superfamily member 9 called 4-1BB (CD137), TNFR superfamily member 4 called OX40 (CD134), or CD27. This helps deliver a second signal that boosts T cells’ function, improves their ability to kill cancer cells, reduces exhaustion, and supports long-lasting immune responses [3]. The schematic representation of CAR structure is presented in Figure 1.

The CAR is composed of three main domains: an ectodomain with a single-chain variable fragment (ScFv) for antigen binding; a transmembrane domain for membrane anchoring; and an endodomain containing co-stimulatory signalling domains (e.g., CD28, 4-1BB, OX40, CD27) and the CD3ζ activation motif.

The concept of a chimeric T-cell receptor (cTCR)—created by fusing antibody-derived variable regions (VH/VL) with constant regions from the T-cell receptor (TCR)—was first introduced in 1987 by Dr. Yoshikazu Kurosawa and his team at the Institute for Comprehensive Medical Science in Aichi, Japan. This ground-breaking study demonstrated that introducing anti-phosphorylcholine chimeric receptors into EL4 mouse T-cell lymphoma cells triggered a calcium response when the cells were exposed to phosphorylcholine-positive bacteria. This finding demonstrated that the chimeric receptors were capable of activating T cells upon recognising specific antigens [5]. Research work on the first-generation CAR T cells lasted from 1989 to 1993. Israeli immunologists Zelig Eshhar and Gideon Gross, working in the Department of Chemical Immunology at the Weizmann Institute of Science in Rehovot, developed the first genetically engineered T cells expressing a CAR [6,7]. These functional cTCRs were successfully expressed on the T-cell surface and demonstrated antigen-specific binding to trinitrophenyl (TNP), leading to T-cell activation, as evidenced by IL-2 secretion and target cell cytotoxicity. Importantly, the activation of cTCR-expressing T lymphocytes occurred independently of MHC restriction, as shown by IL-2 production in response to TNP-conjugated proteins immobilised on plastic substrates [7]. Despite this significant proof of concept, these first-generation CARs lacked clinical efficacy [2,6].

Since then, CARs have evolved from the first generation to the fifth generation, with lower toxicity and better therapeutic effects in patients with lymphoma, leukaemia, and multiple myeloma. While the overall structure of CARs has remained relatively unchanged, the design and function of the intracellular domain have undergone significant modifications across different generations. Second-generation CAR T cells include the CD3ζ domain and co-stimulatory molecules (CMs), such as CD28, CD134 (OX-40), and CD137 (4-1BB), which enable dual signalling. Adding a CM improves T-cell activation, survival, expansion, proliferation, and other antitumour effects. CD137-based CARs tend to last longer in the body but activate more slowly, while CD28-based CARs trigger faster and have stronger responses with better T-cell expansion and memory formation [3]. Second-generation CARs targeting CD19 have been highly effective in treating B-cell cancers. A newer CD19 CAR therapy, obe-cel (CAT-4-1BB-Z), is showing promising results in clinical trials for relapsed B-cell ALL. However, issues like relapse and limited persistence still exist, pushing the development of third-generation CARs [3].

Third-generation CAR cells consist of CD3ζ and various CMs such as CD28, CD137 (4-1BB), CD134 (OX-40), NKG2D, CD27, TLR2, and inducible T-cell co-stimulator (ICOS). Those combining CD3ζ-CD28-4-1BB-based CAR-T cells are the most widely used. However, third-generation CAR T cells have more severe side effects and quicker T-cell exhaustion compared to second-generation CAR T cells. It is primarily due to overstimulation from multiple CM-mediated signals [3,7]. Fourth-generation CAR T cells are also known as TRUCK (T-cell redirected for universal cytokine-mediated killing), UniCAR-T (universal CAR-T), or armoured CAR T cells. These “armoured” CARs enhance T-cell expansion, persistence, and antitumour activity while lowering systemic toxicity. They also help restore the immune system after treatment. Still, they are less effective against solid tumours and may cause side effects from cytokine release in healthy tissues. Fourth-generation CAR T cells are based on second-generation designs but incorporate significant modifications to the intracellular domain. They add a nuclear factor of activated T cells (NFAT)-responsive element that stimulates the production of immune-boosting proteins, such as cytokines (IL-2, IL-12, IFN-γ) and co-stimulatory signals (CD28, OX40, 4-1BB), when the CAR is activated. This helps shape a stronger immune response and improves the tumour microenvironment [3,7].

The fifth generation, also incorporating the characteristics of second-generation CAR-T cells, expands on the endodomain by including an IL-2Rβ membrane receptor. Such an addition enables the cytokine-inducing JAK-STAT3/5 pathway to be activated by supplying a binding site for STAT3. Fifth-generation, or “next-generation”, CAR-T cells are recurrently modified to better answer the challenges of CAR-T-cell therapy [3]. The schematic representation of CAR-T-cell generations is presented in Figure 2.

There is another alternative to CAR-T cells, namely, chimeric antigen receptor natural killer (CAR-NK) cells, which are supported by increasing preclinical evidence. They can be administered without donor–recipient HLA matching, reducing both toxicity and treatment cost. CAR-NK cells have shown efficacy in haematological malignancies, with growing evidence supporting their potential against solid tumours in both preclinical and clinical models [4].

Researchers are working to expand the use of CAR-T-cell therapy in solid tumours. TRUCKs have emerged as an evolution in CAR technology. They release ILs that increase inflammation, modify the tumour microenvironment, and attract additional T cells to penetrate solid cancers. For example, research into the use of CAR-T-cell therapy in glioblastoma has not yielded promising results. Despite intensive multimodal treatment, outcomes for patients with glioblastoma remain unfavourable [8]. Researchers are exploring the potential of CAR macrophages (CAR-Ms) as a new approach to cancer treatment. CAR-Ms offer a promising strategy for targeting solid tumours by genetically modifying human macrophages with specific chimeric antigen receptors (CARs). This enhances their ability to recognise and engulf cancer cells, as well as present tumour antigens more effectively. Essentially, CAR-M therapy involves inserting a tailored CAR gene into macrophages, enabling them to identify tumour-specific antigens, bind to cancer cells, and trigger an immune response against the tumour [9].

Another study has shown that a CAR-M therapy targeting HER-2-positive solid tumours has shown promising results. The study was performed on 19 patients (16 osteosarcomas, 1 primitive neuroectodermal, 1 Ewing sarcoma, and 1 protofibroblastic small round cell tumour). Among the 19 treated patients, the median overall survival was 10.3 months (range: 5.1–29.1 months), with only one case of high fever reported as an adverse event. However, a significant challenge remains. Macrophages are unable to self-proliferate. The advantages of CAR-Ms include intense infiltration into solid tumours, effective phagocytosis, and the ability to present tumour antigens [10,11]. CAR-T therapy is a revolutionary type of immunotherapy that has demonstrated significant success in treating various cancers, particularly blood-related malignancies. However, its effectiveness remains limited when it comes to solid tumours such as colorectal cancer, neuroblastoma, and glioblastoma [7].

## 2. The Tumour Microenvironment (TME) in Solid Tumours

The tumour microenvironment (TME) in solid tumours plays a crucial role in interaction with the immune system, leading to possible progression. It is composed not only of cancerous cells but also endothelial cells and stromal fibroblasts, as well as extracellular matrix components, signalling molecules, and structural features that together create a complex and often immunosuppressive environment [12]. Understanding the components and functions of the TME is essential for identifying improvements in cancer therapies, particularly immunotherapy. The elements of the TME are schematically presented in Figure 3.

### 2.1. What Makes the TME Immunosuppressive?

The immunosuppressive nature of the TME is a major barrier to successful antitumour immunity. Tumours create an environment that actively excludes or disables immune effector cells, particularly cytotoxic T lymphocytes (CTLs) and natural killer (NK) cells, which are essential for recognising and eliminating cancer cells [13]. Apart from cellular mechanisms of resistance, tumours are also capable of utilising soluble factors, as well as immune checkpoint signalling [14]. The most important factors, already confirmed by recent research results, will be discussed here. However, the TME topic is extremely broad and also discussed by others [15,16,17,18,19,20].

#### 2.1.1. Immune Cells

In the TME, several immune cells can be found to suppress antitumour immunity. These include regulatory T Cells (Tregs), which are a subset of CD4+ T cells characterised by the expression of FOXP3(+) and high expression of CD25(+hi) [21]. Tregs accumulate in large numbers and exert potent immunosuppressive effects that impair antitumour immunity. They inhibit the activation and differentiation of CD4+ helper T cells and CD8+ CTLs, suppress antigen presentation by dendritic cells (DCs), and secrete immunosuppressive cytokines, including transforming growth factor beta (TGF-β), IL-10, and IL-35. These cytokines downregulate immune responses and promote the expression of exhaustion markers on intratumoural CD8+ T cells. Tregs also release perforin and granzyme, which leads to the killing of effector immune cells, as well as interference with immune cell metabolism via the secretion of cyclic adenosine monophosphate (cAMP). Their presence in tumours has been strongly associated with poor prognosis and enhanced tumour progression. The transcription factor FoxP3, which has a crucial role in maintaining Treg function, has made it a key target for immunological studies, as its stable expression is required for suppressive activity [22].

In addition to Tregs, tumours also infiltrate myeloid cells, called myeloid-derived suppressor cells (MDSCs) [21]. They are an essential part of innate immunity, contributing to pathogen clearance, antigen presentation, and inflammation. Under physiological conditions, myelopoiesis leads to the differentiation of progenitor cells into mature monocytes, granulocytes, and dendritic cells [23]. However, this process becomes dysregulated in cancer and other chronic inflammatory conditions. Inflammatory signals impair the maturation of immature myeloid cells (IMCs), resulting in the accumulation of a heterogeneous population with potent immunosuppressive activity [24]. MDSCs suppress antitumour immunity through multiple mechanisms: they express immune checkpoint molecules, deplete essential metabolites such as L-arginine, produce reactive oxygen and nitrogen species, and generate immunosuppressive adenosine [25]. These actions inhibit T-cell activation, proliferation, and function. Additionally, MDSCs contribute to tumour progression by promoting angiogenesis and establishing pre-metastatic niches. This results in the suppression of immunotherapy effectiveness. Thus, they should be considered as important targets for therapeutic interventions aimed at restoring antitumour immune responses [26]. Further key components of the TME are tumour-associated macrophages (TAMs). They enhance tumour cell survival and facilitate invasion by secreting cytokines, growth factors, and inflammatory mediators that remodel the TME, which results in the promotion of immunosuppression, angiogenesis, metastasis, and chemoresistance [27]. Often localised in stromal and perivascular areas, TAMs are especially active in hypoxic and avascular regions, where they drive angiogenesis. Their high plasticity allows them to switch between pro-inflammatory (M1-like) and immunosuppressive (M2-like) phenotypes in response to local signals [16]. Epigenetic mechanisms regulating TAM polarisation are being explored as therapeutic targets, with microRNAs and pharmacological agents under investigation to modulate their function. Furthermore, natural compounds and phytomedicines are being evaluated as adjunct therapies to overcome TAM-induced chemoresistance, offering potential interaction with chemotherapy, anti-angiogenic drugs, and immune checkpoint inhibitors. Targeting TAMs through multi-omics approaches hold promise for advancing personalised cancer therapies [28]. Another significant contribution to TME immunosuppression is made by stromal cells, particularly CAFs, which produce a wide range of cytokines and chemokines that attract immunosuppressive cells such as MDSCs and Tregs. They also secrete ECM components, including collagen and fibronectin, which alter the physical structure of the TME, creating barriers against immune cell infiltration [29]. Additionally, CAFs express immune checkpoint ligands and modulate local metabolic conditions, such as oxygen tension and pH, thereby establishing a pro-inflammatory environment, with decreased immune responses and elevated oxygen levels [30].

Another group of immune cells contributing to immune suppression are tumour endothelial cells (TECs). In response to vascular endothelial growth factor (VEGF), they selectively express adhesion molecules, leading to abnormal vessel formation. The vessels impede the efficient flow of immune cells and support a hypoxic microenvironment, further promoting immune tolerance [31].

#### 2.1.2. Soluble Factors

Not only immune cells take part in developing immune suppression in the TME. Further components are soluble factors, including growth factors such as epidermal growth factor (EGF), VEGF, and platelet-derived growth factor (PDGF), which play a vital role in enhancing angiogenesis, tumour cell proliferation, and metastasis. Elevated levels of VEGF are particularly associated with poor cancer prognosis, making it another important therapeutic target. EGF, on the other hand, plays a role in tumour progression by enhancing cell proliferation, survival, and migration, as it binds to the epidermal growth factor receptor (EGFR) [32]. Aberrant EGFR signalling is frequently observed in various malignancies and is often linked to aggressive tumour behaviour and resistance to therapy [33]. Another factor—PDGF—supports the structural and functional integrity of the TME by promoting the recruitment and activation of stromal cells, including fibroblasts and pericytes [34]. It also promotes angiogenesis, contributing to tumour cell invasion and metastasis [35].

Another particularly well-studied factor is transforming growth factor beta (TGF-β). It is produced by tumour cells, stromal fibroblasts, and cancer-associated fibroblasts (CAFs) [36,37]. TGF-β inhibits the activation of CTLs and NK cells but also suppresses the expression of transcription factors; furthermore, it promotes the conversion of naive CD4+ cells into FOXP3+ Tregs and redirects myeloid differentiation toward MDSCs [38,39]. It also enhances ECM remodelling and creates a unique environment favouring tolerogenic pathways by reprogramming stromal and immune cells [40]. Other key immunosuppressive cytokines include interleukin-10 (IL-10) and interleukin-35 (IL-35). IL-10 is secreted by regulatory cells and TAMs; it inhibits CD8+ T-cell and NK-cell cytotoxicity and suppresses antigen presentation via dendritic cells [41]. IL-35, a recently discovered immunosuppressive cytokine produced by Tregs and regulatory B cells, blocks Th1 and Th17 differentiation and limits early T-cell proliferation, thereby contributing to tumour immune escape [42,43].

The interplay between cytokines, soluble mediators, and cellular components in the TME is complex and crucial for tumour progression. These mediators build up the TME’s structure and immune dynamics and are essential for cancer’s ability to suppress immune surveillance, emphasising their utility as a potential aim in cancer treatment [44]. Also, the cytokine network within the TME plays a crucial role in cancer pathology and the immune response to tumours. Cytokines work as key regulators of both tumour progression and immune suppression. They have the ability to act either as pro-inflammatory or anti-inflammatory signals that influence tumour growth and immune evasion [45,46,47].

### 2.2. Immune Checkpoint Signalling

Immune checkpoint signalling is a crucial mechanism by which solid tumours evade immune surveillance, hindering CAR-T-cell function within the TME. They downregulate T-cell activation pathways, which leads to T-cell exhaustion, reduced cytokine production, and impaired cytotoxic function [48,49]. Their interaction with corresponding ligands—commonly expressed on tumour and stromal cells—leads to T-cell exhaustion, reduced cytokine production, and impaired cytotoxic function. Programmed cell death protein 1/programmed death ligand 1 (PD-1/PD-L1) signalling, in particular, has been shown to diminish the efficacy of CAR-T cells by limiting their proliferation and persistence. Strategies to overcome this type of immunosuppression in the TME include genetic modification of checkpoint receptors in CAR-T cells, co-expression of dominant-negative receptors, or combination therapies including immune checkpoint inhibitors. These approaches aim to restore T-cell activity and sustain antitumour responses in solid tumour settings. Other immune checkpoints, such as cytotoxic T-lymphocyte-associated protein 4 (CTLA-4), T-cell immunoglobulin and mucin-domain containing-3 (TIM-3), lymphocyte activation gene-3 (LAG-3), and T-cell immunoreceptor with Ig and ITIM domains (TIGIT), are also associated with immune suppression by mediating T-cell dysfunction and exhaustion. Tumours that present the co-expression of multiple ligands for these receptors often demonstrate higher resistance to immunotherapy [50]. Emerging data suggest that the upregulation of these alternative checkpoints may be a compensatory mechanism following PD-1 blockade, contributing to resistance to monotherapy [51]. Thus, the integrated blockade of immune checkpoints is being explored to enhance therapeutic efficacy [52].

### 2.3. Tumour-Associated Antigens (TAAs)

Tumour-associated antigens are mutated proteins that are abnormally expressed in tumours, and their presence can contribute as another part of the tumour’s immunosuppression [53]. Chronic exposure of immune cells to TAAs can result in T-cell unresponsiveness or deletion, and some tumours downregulate antigen presentation, leading to immune escape [54].

### 2.4. Physical and Chemical Barriers

The TME also presents physical and metabolic obstacles. Hypoxia leads to the induction of hypoxia-inducible factor (HIF)-1α-driven gene expression, which supports immune tolerance and angiogenesis [55]. Low pH and high lactate levels inhibit T-cell receptor signalling [56]. Nutrient competition between tumour and immune cells for glucose, amino acids, and lipids decreases immune function [57]. Elevated intertumoural fluid pressure and an increased level of extracellular matrix (ECM) stiffness are physical barriers that reduce immune cell infiltration into tumour tissues. On the other hand, the biochemical properties of the extracellular matrix influence immune cell activity, frequently enhancing immunosuppressive phenotypes [58].

### 2.5. Extracellular Matrix

Apart from its structural role, the ECM serves an important role in regulating immune responses within the TME [59]. The extracellular matrix forms a physical barrier due to its density. It reduces the infiltration and migration of effector immune cells such as T cells and natural killer cells [60]. Additionally, cytokines and growth factors retained within the ECM play a significant role in creating an immunosuppressive microenvironment. Enzymes responsible for ECM remodelling, especially matrix metalloproteinases, not only promote tumour invasion and metastasis but also simultaneously influence the phenotype and function of infiltrating immune cells [61].

## 3. Why CAR-T Cells over Traditional Chemotherapy or Radiotherapy?

Throughout the last century, cancer therapy has been improved and evolved into more effective treatments. Two components of traditional treatment have been the basis of the majority of cancer regimens: radiotherapy and chemotherapy [62]. The former, first used in the early 1900s, is now used in over 50% of cancer patients, primarily in stereotactic radiation therapy, image-guided radiation therapy, and adaptive radiation therapy [63,64,65]. The latter, since its introduction in the 1950s, is commonly used as part of combination treatments [66]. Both, however, have their drawbacks, in the form of toxicities, tumour resistance, or malignancies.

### 3.1. Toxicities of Chemotherapy

Conventional chemotherapy displays vast side effects, ranging from hypersensitivity reactions to system-specific toxicities. Different classes of drugs are often associated with particular symptoms, albeit it is not the point of this paper to discuss them in depth.

Nausea and emesis are some of the most common side effects, experienced by 40% to 96% of patients undergoing treatment. This not only puts the patient in severe psychological discomfort but also leaves them at risk of malnutrition and acid–base imbalance [67,68]. Other major gastrointestinal symptoms include diarrhoea and mucositis, which can, in turn, lead to dose reduction and impair treatment effectiveness [69].

Another dose-limiting agent—cardiotoxicity—can manifest both in the short and long term post chemotherapy as congestive heart failure, hypertension, rhythm abnormalities, and cardiomyopathy. Chronic cardiotoxicity associated with anthracycline presents in up to 57% of patients (although more studies are needed) [70]. Heart-related complications reign as one of the greatest mortality causes in cancer survivors [71,72].

Since most cancer drugs are excreted with urine, the kidneys are exposed to their toxic effects. Therapeutics can cause intratubular obstruction due to crystal depositions, accumulate in cells, or damage and interfere with DNA repair. This results in a decreased glomerular filtration rate, acute tubular necrosis, acute kidney injury, and possibly chronic renal failure [71]. Currently, the treatment of chemotherapy-induced chronic renal failure is limited [73].

Other toxicities of chemotherapy are presented in Table 1.

### 3.2. Toxicities of Radiotherapy

Despite modernisation and increasing precision of radiation therapy, higher doses and combination treatments result in an array of clinically significant side effects. The following sections will discuss some of these toxicities.

A 2022 retrospective study analysing 181 breast cancer patients treated with different types of axillary radiotherapy shows that, overall, 20.4% of these women developed lymphedema. It was more common in patients treated with conventional radiotherapy (42.2%), whilst in hypofractionated treatment, the rate was significantly lower (8.5%) [77]. Lymphedema manifests with pain, oedema, and restricted arm movement and can be either acute or late [78].

Thoracic radiotherapy is also supposedly linked to cardiotoxicity. Studies show that not only is a higher dose of radiation proportional to the occurrence of cardiac problems but low recurrent dosages also pose a risk factor [78]. Long-term cancer survivors are the most affected. Valvular heart disease develops in around 60% of patients 20 years post exposure [79]. The mechanism of impairment could be direct damage and radiation-induced inflammation, ischemia, and fibrosis [80].

### 3.3. Toxicities of CAR-T-Cell Therapy

The aforementioned glossary of traditional cancer therapeutics’ toxicities serves as a basis for comparison to CAR-T cells. The most common adverse effect of CAR-T therapy is cytokine release syndrome (CRS), which occurs in between 50 and 90% of treated patients. There are different CRS grading systems; however, as per the 2018 ASTCT consensus backed by the American Society for Blood and Marrow Transfusion, it is divided into 4 grades. Grade 1 CRS, considered mild, presents with flu-like symptoms that develop within a few weeks of administration, typically characterised by fever over 38 °C that is not attributed to any other cause. Grade 2 is additionally accompanied by hypotension, which can be managed with fluids, without the use of vasopressors, and hypoxia managed with a low-flow nasal cannula. In grade 3 CRS, hypotension requires a vasopressor, either with or without vasopressin, and hypoxia is managed with more intense oxygen therapy, for example, using a high-flow nasal cannula. For CRS to be considered grade 3, the occurrence of one out of two aforementioned severe events is enough. Grade 4 CRS requires multiple vasopressors (without vasopressin) and positive-pressure oxygen therapy [81]. It comprises systemic inflammation, significant hypoxia, shock, and organ toxicities. Every patient demonstrating signs of CRS should be closely monitored [82,83]. Facilities providing CAR-T-cell therapy are advised to train their staff in the early recognition of CRS symptoms [84]. The guidelines presented by the Society of Immunotherapy of Cancer (SITC) recommend admission of patients treated in outpatient facilities even at the early stages of CRS. In severe CRS, treatment guidelines recommend Tocilizumab, a monoclonal antibody that targets IL-6 receptors. It was approved by the FDA in 2017 for the treatment of CRS in patients aged 2 years and older [85]. In the FDA’s retrospective analysis of clinical trials, Tocilizumab demonstrated a 69% response rate in a group of 45 patients, with the following premises for CRS resolution: resolution within 14 days, administration of no more than two doses of Tocilizumab, and administration of no other medication except glucocorticosteroids. Prophylactic use of Tocilizumab is still being discussed. In a 2021 clinical study, 10 out of 20 patients treated with prophylactic Tocilizumab did not develop CRS, and there were no cases of grade 3 CRS. Out of the other 10 patients, 7 developed grade 1 CRS, and 3 developed grade 2. There were no adverse effects associated with Tocilizumab [86]. The SITC guidelines do not yet recommend the use of Tocilizumab in all patients receiving CAR-T; however, they do suggest it for at-risk patients [85]. Although the sample size is small, similar and extended studies could spark further conversation about the prophylactic use of Tocilizumab in all patients receiving CAR-T-cell therapy.

Immune effector cell neurotoxicity syndrome (ICANS) is another toxicity that manifests in patients, oftentimes post experiencing CRS. Neurological symptoms appear between 1 and 3 weeks after the infusion, typically as aphasia, dysphasia, language difficulties, seizures, and confusion. It lasts anywhere from 5 to 11 days and is closely linked to CD19 CAR-T-cell therapy [87]. In a 2024 preclinical study, researchers engineered CAR-T cells to self-regulate cytokine release using Toci, a Tocilizumab-based single-chain variable fragment. When used in humanised mice, these cells demonstrated a reduction in CAR-T-cell-induced side effects. The findings show promise for improving the management of CRS and ICAN when used in combination with Tocilizumab [88].

Side by side, there is no way to categorically describe the side effects of CAR-T therapy as “better” or even “milder”. They are different, and that can be an advantage. They are more acute, which makes them easier to detect and treat efficiently. CRS is preceded by fever, 1 to 3 days before the onset of other symptoms [83], and with properly trained medical personnel and adequately equipped institutions, it can be spotted early and treated before its further escalation. Importantly, most of these toxicities are reversible and rarely lead to chronic organ damage presented in traditional treatments, which could potentially improve patients’ quality of life in the long term [87].

### 3.4. Short Treatment Duration and Potential for Long-Term Remission

The process of administering CAR-T-cell therapy consists of several stages: patient evaluation and selection, leukapheresis, cell engineering, lymphodepletion, administration, and post-infusion monitoring [89]. Leukapheresis does not require long hospitalisation, and the median time of the procedure is 240 min [90]. The manufacturing of CAR-Ts takes about 1 to 2 weeks [89]. In the ELIANA and ENSIGN clinical trials, the median time from enrolment to infusion of the cells was 43 days [91]. In a study conducted at the University of Vienna, this period averaged 69 days [92]. The average hospitalisation time for patients with different haematological malignancies is 12 days from the time of infusion, according to data collected in the US by the American Society of Clinical Oncology [93]. These timelines vary based on the type of cancer, patients’ overall well-being, manufacturing process, and facilities conducting the treatment. Nevertheless, if appropriately managed, the treatment time is approximately 2 to 3 months, during which the actual time spent in the hospital is 1 day for harvesting the cells and a few weeks for the infusion, monitoring, and handling possible side effects.

Currently, CAR-T-cell therapy is considered to have a good initial response rate, with a notable improvement around the 3- or 6-month mark in various haematological malignancies [94,95,96]. However, the most important part of CAR-Ts is their potential for long-term remissions. In a follow-up study of the LEGEND-2 trial, the progression-free survival rate was 21%, and the overall survival rate was 51%, among the 74 enrolled paediatric patients [97]. In another trial, two out of three patients receiving CD-20 CAR-T treatment maintained a 7-year remission before relapsing [98]. Durable responses were also noted in the case of follicular lymphoma, with complete remission lasting over 9 years, followed closely by chronic lymphocytic leukaemia (CLL) and diffuse large B-cell lymphoma (DLBCL), with a duration of over 8 years [99]. Whilst these results are rare and the data on the late efficacy of CAR-Ts are sparse and require more research, they show that there is a real possibility of achieving them. Strategies proposed to enhance CAR-T persistence are optimisation of culture conditions, combination therapy with epigenetics, PI3K-AKT or other medication (ex., metformin), use of metabolites that enhance oxidative phosphorylation (OXPHO) and reduce glycolysis, and genetic manipulation with the clustered regularly interspaced short palindromic repeat (CRISPR) system [95]. Data also suggest that a shorter time between leukapheresis and infusion has a positive impact on complete response and overall survival rates [100,101].

The following table concludes most common toxicities of each therapy discussed in previous sections.

## 4. Engineering CAR-T Cells to Overcome Immunosuppression

The immune system faces suppression through various mechanisms, which tumours use to control effector T-cell activity by employing Tregs, as well as inhibitory cytokines (TGF-β, IL-10) and immune checkpoint ligands [102]. Standard CAR-T products achieve high success rates in haematologic cancers; however, their limited ability to infiltrate and persist in solid tumours leads to suboptimal treatment outcomes [103].

A significant obstacle for CAR-T-cell therapy in solid tumours is understanding the correlation between the TME and a positive outcome of the treatment. It is now recognised that the persistence and therapeutic efficacy of CAR-T cells are hindered by physical barriers and components of the TME, including stromal and immune cells, which are involved in secretion of various immunosuppressive factors. Accordingly, it has been demonstrated that remodelling the TME or conferring intrinsic CAR-T-cell resistance to immunosuppression may provide a more effective therapeutic strategy than single-pathway targeting [104].

Scientists employ gene engineering techniques to address these obstacles. The approaches for gene engineering can be classified into four main categories: checkpoint blockade integration, metabolic adaptation, targeting suppressive cells or molecules, and cytokine and receptor armouring [105].

Disabling inhibitory checkpoint receptors on CAR-T cells enhances their function in the solid TME. CRISPR/Cas9-mediated PD-1 knockout in CAR-T cells has shown improved persistence and cytotoxicity against solid tumours [106]. Co-delivering PD-1-blocking single-chain antibodies through a bicistronic CAR backbone similarly enhances antitumour activity while also minimising systemic toxicity [107]. However, this topic warrants further research [108].

Another strategy used to overcome immunosuppression in the solid tumour microenvironment is metabolic reprogramming. One of the factors that influences the reduced response to treatment is inadequate vascularisation of the tumour and, consequently, hypoxia. Hypoxia negatively affects the distribution of drugs to tumour masses [20]. The factors that lead to hypoxia are the rapid and uncontrolled multiplication of cancer cells and abnormal vascularisation. Insufficient oxygenation can affect the immune response against cancer [104].

The concentration of oxygen affects several aspects of therapy. The differences and possibilities of designing CAR-T cells that change their properties and the nature of their action depending on the environment in which they are located should be taken into account. Designing the cells in such a way that they are inactive in an environment with higher oxygen levels and more active at lower oxygen concentrations would limit the harmful effect on healthy tissues. This system uses the endogenous way of detecting oxygen by T cells. It’s achieved either by inserting hypoxia-responsive element (HRE) sequence into the promoter or by combining the hypoxia-inducible factor domains of hypoxia-inducible factor (HIF) with the intracellular domain of CAR. This procedure allows for hydroxylation and degradation in higher oxygen concentrations. An equally interesting way to influence the TME is to direct T cells against antigens that are overexpressed in conditions of insufficient oxygenation [104].

Also, CAR T cells can be gifted with cytokine signals or decoy receptors to overcome local suppression. IL-7 or IL-18 secretion by CAR T cells supports expansion of effector function within nutrient-poor TMEs [1].

Furthermore, directly eliminating or reprogramming immunosuppressive elements can enhance CAR-T efficacy [102]. CAR-T constructs targeting macrophage-associated antigens, for example, CD123, exhaust tumour-associated macrophages, alleviating myeloid-derived suppression [104]. As an alternative, bispecific CAR-T cells that recognise both tumour antigens and Treg markers, can achieve double targeting without need for exogenous agents [109].

Due to the complexity of the immune system, CAR therapy can act on multiple different levels, and its effects on TAMs, MDSCs, and Tregs are particularly important. One of the modifications applied to the therapy is the introduction of the folate receptor FRβ, which is used to target TAMs and MDSCs. Therefore, using FRβ-CAR-T in the tumour microenvironment eliminates TAMs. Additional effects on TAMs with a tyrosine kinase receptor can be induced by targeting CAR-T cells to the growth-arrest-specific protein 6 GAS6. In addition, inhibition of sialic acid-binding immunoglobulin-like lectin 1CD47/SIRPα and CD24/SIGLEC-10 signalling enhances the mechanism of phagocytosis. The effect of CD24–CAR-T cells is the polarisation of macrophages towards the M1 phenotype, counteracting immunosuppression induced by the M2-type TME. Polarisation is also supported by IFN-γ and TNFα, which further reduces the immunosuppressive effect [20].

It is also worth mentioning that the cells contributing to the increase in the population of TAMs and MDSCs are TREM2+ cells. They have an analogous effect to the above-described M2-type TAMs and block the proliferation of T lymphocytes. The receptor associated with TNF-TRAIL ligand-induced apoptosis TR2, which is located on MDSCs and activates apoptosis after binding to the TRAIL ligand, can be used. For this purpose, a fragment encoding an antibody antagonist to TR2, specifically an scFv, is utilised [20].

For now, engineering CAR-T cells to resist immunosuppression has unlocked new therapeutic avenues for solid tumours. There are numerous preclinical trials aiming to increase the efficacy of CAR-T in solid tumours. Future work should optimise combinatorial armouring, such as multiplexed genes [74], improve trafficking via chemokine receptor co-expression, and refine safety switches to mitigate off-tumour effects. The integration of synthetic biology tools, including logic-gated CARs and drug-inducible systems, will provide additional precision [75]. Thus, as research evolves, armoured CAR-T therapies may overcome current limitations and deliver durable remissions across diverse cancers.

## 5. Challenges Faced by CAR-T Cells in Solid Tumours

For CAR-T-cell therapy to work, the CAR-T cells must target tumour cells without interacting negatively with other healthy cells. Solid tumours tend to express TAAs, which multiple cancer treatment approaches can target. However, despite being mostly expressed by tumour cells, they are sometimes found in normal healthy cells [110].

It has been extraordinarily troublesome to find a protein appearing specifically in a tumour’s structure but at the same time not being a part of a healthy cell. Finding such proteins would certainly revolutionise cancer treatment. A promising example of this has been the identification of follicle-stimulating hormone receptors (FSHRs) expressed only in ovaries [111]. The development of a tumour in a woman’s organ other than the ovary results in blood vessels of that tumour expressing FSHRs [111]. Due to this fact, it is possible to target the cells of specific tumours’ blood vessels with CAR-T-cell therapy and eliminate them, resulting in an impaired nourishment factor and significantly improved cancer remission chances.

Another key challenge in CAR-T-cell therapy is finding the most effective way to deliver these cells to the intended location of a tumour. Deciding whether to navigate CAR-T cells through blood vessels or distribute them locally is crucial to achieve the best possible outcome for the patient. Cancer cells tend to trick the host immune system and continually suppress T cells’ functions, reducing their therapeutic effectiveness [48,112].

Research carried out on an orthotopic model mimicking human pleural malignancy showed that M28z CAR-T cells delivered intrapleurally significantly outperformed the same cells that had been systemically infused. The intrapleural route of delivery required 30 times fewer CAR-T cells to cause long-term remissions [113]. Additionally, intrapleural CAR-T-cell administration promoted the elimination of extrathoracic tumour sites as a result of early CD4(+) T-cell activation and T-cell-mediated cytotoxicity [113]. CAR-T-cell therapy delivery will be further discussed at length in Section 6.

Implementing a 1:1 ratio of CD4+ CAR-T cells to CD8+ CAR-T cells revealed improved antitumour efficacy in leukaemia and lymphoma therapy [114]. It may be connected to IFN-γ upregulation enhancing the therapeutic effect [114]. The release of IFN-γ by CD4+ CAR-T cells increases cytotoxic reactions directed towards solid tumour cells [114]. On the other hand, CD8+ CAR-T cells induce a release of granzyme and perforin in solid tumour cells, which contributes to the synergistic activity of both types of cells [114]. Additionally, incorporating natural killer T (NKT) cells enhances the safety and efficacy of treatment in solid tumours, mainly through the activation of apoptosis pathways caused by the release of perforin and granzyme B levels [114]. To further enhance the therapeutic effect, IL-15 may be implemented, which is essential for T-cell memory and also helps in developing and maintaining stable, appropriate conditions for NKT cells (e.g., protection from hypoxia) [114]. Another key factor in CD4+ and CD8+ therapy is monitoring the tumour T-cell infiltration. In patients with large early-stage cervical cancer, a higher CD8/CD4 ratio, as well as a higher CD8/Treg ratio, is associated with the absence of lymph node metastases [115]. The normal CD4+ and CD8+ cell ratio in a healthy adult is approximately 2:1; however, the effectiveness of ratios other than 1:1 in therapy is yet to be discovered. Perhaps delivering these cells using a predefined time interval and order would result in increased therapy efficacy. Current challenges and findings in the CAR-T strategy are presented in Table 2.

## 6. Innovative Delivery Approaches

### 6.1. Administration Methods

While intravenous administration remains the standard route for approved CAR-T-cell therapies, its efficiency in solid tumours is limited due to challenges with trafficking and off-target toxicities [120]. Local administration strategies, such as intrapleural, intraventricular, and intracavitary injections, offer a significant advantage by evading the need for CAR-T cells to navigate complex tumour vasculature and stromal barriers, thereby reducing systemic toxicity by limiting CAR-T cells’ exposure to healthy tissues [1]. On those tissues, many CAR-T-cell targets for solid tumours are present at low levels, contributing to on-target off-tumour toxicity [1,120]. For example, mesothelin, CEACAM5, and ERBB2 are expressed on benign lung epithelial cells, resulting in toxic effects on pulmonary tissue [120]. Thus, local delivery of CAR-T cells may prove beneficial by reducing those harmful effects [120].

Preclinical studies on murine models have demonstrated that the intraventricular delivery of CAR-T cells is highly efficient in targeting human epidermal growth factor receptor 2 (HER2) and interleukin-13 receptor subunit alpha-2 (IL13Rα2) in breast cancer brain metastases and glioblastoma [1]. Later clinical trials have further investigated various administration methods targeting these proteins, with significant promise shown in IL13Rα2-targeted CAR-T-cell therapy [121]. Similarly, the intrapleural administration of mesothelin-directed (MSLN) CAR-T cells in malignant pleural mesothelioma showed safety and antitumour activity during clinical trials [120]. Localised CAR-T-cell delivery is also under investigation for ovarian and pancreatic cancers, reflecting growing interest in this strategy [120]. However, the effectiveness of this approach appears to be limited to the treatment of single lesions or oligometastatic disease [1].

Innovative methods, such as the encapsulation of CAR-T cells in fibrin gels, have shown excellent outcomes compared to direct injection or intravenous injection in murine models of unresectable adenocarcinoma and glioblastoma [120]. This delivery system enhances the persistence and function of CAR-T cells while potentially alleviating on-target off-tumour toxicities, as it allows localised, sustained release within the tumour bed [120]. While hopeful, the results of preclinical studies on small animal models may not translate to a successful treatment for human patients.

The intracranial postsurgical delivery of CAR-T cells has also demonstrated therapeutic promise in multiple clinical studies [120]. Trials involving CARv3-TEAM T cells targeting EGFRvIII, infused intraventricularly, and bivalent CARs targeting EGFR and IL13Rα2 via the intrathecal route have shown rapid and significant radiographic tumour regression, although some patients experienced early-onset neurotoxic complications that were managed with steroids and IL-1R antagonists [120]. A landmark case demonstrated a complete response in recurrent multifocal glioblastoma during a clinical trial using multiple intracavitary and intraventricular infusions of IL13Rα2-specific CAR-T cells, highlighting the viability of repeated locoregional treatment [122].

### 6.2. CAR-T-Cell Engineering for Enhanced Penetration

The therapeutic success of CAR-T cells in solid tumours is hindered by their limited ability to traffic to and infiltrate tumour tissue due to barriers expressed by the immunosuppressive TME (e.g., ECM components, rich tumour stroma, hypoxia, nutrient deprivation, low pH), as well as heterogeneous tumour antigen expression, antigen loss, immunosuppressive cytokines, fibrotic tumour stroma, and abnormal vasculature—all expanded on in Section 2 [1,122].

To address these challenges, CAR-T cells that are being engineered to express heparinase exhibit a significantly improved ability to penetrate tumour stroma and enhance antitumour effects due to their ability to degrade heparan sulfate proteoglycans (HSPGs)—one of the ECM components [1]. Targeting fibroblast activation protein (FAP) on CAFs has been shown to diminish the immunosuppressive stroma and inhibit tumour growth [1]. However, FAP-targeted CARs may also affect multipotent bone marrow stromal cells, leading to toxicities such as cachexia and bone damage in some models [122].

CAR-T cells may also be altered to specifically target tumour vasculature, such as VEGFR2, αvβ3 integrin, or PSMA, disrupting blood supply and enabling better T-cell access [122]. Combining CAR-T therapy with anti-angiogenic agents or endothelial-targeted immunotoxins further enhances this effect and can prevent vascular-mediated immune evasion [122]. CAR-T-cell engineering is expanded on in Section 4.

## 7. Combination Therapies

The combination of CAR-T therapy with additional treatments serves as a fundamental approach to defeat the immunosuppressive challenges that solid tumours present. Research in the early stages showed that immune checkpoint inhibitors like anti-PD-1 or anti-PD-L1 antibodies can restore CAR-T-cell function, which results in better cell proliferation and persistence and stronger antitumour effects [107]. Oncolytic viruses (OVs) work together with CAR-T cells to destroy tumour cells, which releases new antigens and creates an inflammatory environment that enhances T-cell movement and activation in areas that were previously tumour-resistant [89]. The administration of intratumoural OVs before CAR-T infusion yields better tumour management and extended survival outcomes than single-agent treatments across various solid tumour models [123,124].

Simultaneously, low-dose chemotherapy and focal radiotherapy have been repurposed to prepare the tumour environment for CAR-T engraftment. Administration of lymphodepleting agents such as cyclophosphamide or fludarabine, prior to CAR-T transferral, depletes regulatory T cells and myeloid-derived suppressor cells, thereby clearing niches for CAR-T expansion [125]. Clinical trials that paired epidermal growth factor receptor variant III (EGFRvIII)-targeted CAR-T cells and radiotherapy, on the other hand, showed synergistic effects, demonstrating the translational potential of this approach [126].

Targeting metabolic checkpoints in the tumour microenvironment represents another point of combination therapy. Small-molecule inhibitors of indoleamine-2,3-dioxygenase (IDO) and adenosine A2A receptors can reprogram the metabolic environment, mitigating kynurenine-mediated suppression and adenosine-driven T-cell anergy, which sustains CAR-T effector function under nutrient stress [127]. Recent work shows that overexpression of adenosine deaminase in CAR-T cells converts immunosuppressive adenosine into inosine and has shown improved functionality. It suggests a novel approach to enhance CAR-T function [128].

Vaccination strategies have been integrated to boost both engineered and endogenous immunity. Amplifying RNA vaccines co-delivered with CAR-T cells shows in early trials that it enhances CAR-T expansion, promotes epitope spreading, and also improves responses in patients with relapsed or refractory solid tumours without adding toxicity [129,130]. mRNA-based boosters encoding target antigens further extend these benefits by recruiting polyclonal T-cell populations that complement CAR-T-mediated cytotoxicity and mitigate antigen escape [131].

In the developing field, combination therapies also include integration with more focused modalities like augmenting bispecific CAR-T constructs or co-administration with tyrosine kinase inhibitors like dasatinib, which has the ability to rest CAR-T cells, reverses exhaustion, and enhances their long-term efficacy [132]. CAR-T designs with synthetic Notch receptors, especially those utilising logic gates, possess a unique capability of providing precise conditional activation, enabling these cells to differentiate between cancerous and normal tissues under challenging scenarios [133].

In combination therapies, the combinatorial approaches summarized in Table 3, including immune modulation with checkpoint inhibitors and other OVs as well as conditioning the tumour microenvironment with chemotherapy, radiotherapy, metabolic inhibitors or vaccines, and targeted agents have a common goal of disrupting the intricate immunosuppressive architecture of solid tumours. An ongoing fusion of systems in immunology, active patient stratification using biomarkers, and engineering methods will be critical in these multi-faceted strategies enabling meaningful remission in patients with solid tumours.

Table 3 summarises the mentioned key approaches, mechanisms of action, and benefits in overcoming suppression encountered in solid tumour microenvironments.

## 8. Clinical Advances and Emerging Data

Despite the significant success of CAR-T-cell therapy in the treatment of haematological malignancies, its effectiveness in solid tumours remains markedly limited due to factors such as antigen heterogeneity and the immunosuppressive nature of the tumour microenvironment (TME) [1,118,136].

A major challenge in applying this therapy to solid tumours is the lack of an appropriate, tumour-specific antigen that can be safely and effectively targeted. Unlike haematological malignancies, which typically exhibit homogeneous and specific surface markers, solid tumours are characterised by significant antigenic heterogeneity. Moreover, they often share the expression of specific antigens with healthy tissues, which significantly hinders the accurate targeting and elimination of all cancer cells and increases the risk of severe toxicity due to the destruction of healthy cells [116].

In recent years, the potential efficacy of CAR-T-cell therapy against solid tumours has been demonstrated. In the treatment of *HER2*-positive gastric cancer (GC), CAR-T cells engineered to target this antigen effectively eliminated *HER2*-positive tumour cells derived from patients, resulting in improved therapeutic outcomes and prognosis. These constructs were second-generation CARs. Studies confirmed that *HER2*-specific CAR-T cells recognise tumour cells in an MHC-independent manner and induce their apoptosis. In *HER2*-positive xenograft tumour models, CAR-T therapy showed strong antitumour activity and high cytotoxicity compared to control groups. Furthermore, it was demonstrated that CAR-T cells exhibit greater precision and effectiveness than unmodified T lymphocytes [117].

Another study evaluated the safety and clinical efficacy of CAR-T-cell therapy in patients with advanced tumours exhibiting *HER2* overexpression. One patient achieved a partial response lasting 4.5 months, while five patients experienced disease stabilisation with a median progression-free survival of 4.8 months. These findings suggest that CAR-T therapy may offer significant clinical effectiveness. However, it is essential to note that one patient with advanced *HER2*-positive gastric cancer developed severe upper gastrointestinal bleeding, indicating that this therapy may carry certain risks during tumour eradication [117].

Currently, numerous studies are being conducted on CAR-T-cell therapy for the treatment of solid tumours. One such study evaluates CAR-T-cell therapy targeting *EGFR*. In a phase I trial involving patients with NSCLC, fourteen patients were enrolled. A partial response lasting from two to four months was observed in four individuals, while stable disease was seen in eight patients. Although grade 3 adverse events occurred, they were manageable. The CAR constructs used in this trial were second-generation and did not contain any tumour-microenvironment-targeting elements. The median progression-free survival (PFS) was three months, and the median overall survival (OS) was 4.9 months [118].

In another phase I study involving patients with relapsed/refractory non-small-cell lung cancer (NSCLC), a therapeutic response was observed. One of nine patients achieved a partial response that persisted for over thirteen months. Stable disease was reported in six patients, while disease progression was noted in two cases. Consistent with earlier trials, second-generation CAR-T cells were used. All adverse events of grades 1–3 were manageable with appropriate treatment. The median PFS was 7.13 months, and the median overall survival (OS) was 15.63 months [118].

It is important to emphasise that the effective application of CAR-T-cell therapy in the treatment of solid tumours still remains a major challenge. So far, no significant breakthroughs have been achieved, which presents a difficulty but also an opportunity [119]. One must believe that with ongoing technological advancements, the use of CAR-T therapy will eventually become a standard treatment option for patients with solid tumours.

## 9. Future Directions

### 9.1. IL-12- and IL-18-Secreting CAR-T Cells

Attempts to treat solid tumours with redirected T cells have so far been unsuccessful. A major challenge in CAR-T-cell therapy in solid tumours is the limited efficacy related to the tumour microenvironment, poor infiltration, and exhaustion of T-cell effector function. In one of the preclinical studies, it was therefore decided to investigate whether reprogramming T cells towards a more acute response would improve immune control of tumours. IL-18 was identified as a potent cytokine that induces T-bethigh FoxO1low in CAR-stimulated T cells. However, it has previously been assumed that increased expression of T-bet factor and decreased FoxO1 are a signature of an acute inflammatory response. This can lead to a more intense immune response against well-developed tumours. This approach was successful in immunocompetent mice with advanced pancreatic adenocarcinoma [137]. In yet another preclinical study, IL-12 and IL-18 normally produced by myeloid cells were shown to stimulate CAR-T-cell activation and could alter the TME to a less immunosuppressive state. In the study, CAR-T cells were reactive with mouse delta-like protein 3 (DLL3), allowing for the examination of CAR-T-secreted IL-12 and IL-18 in an mSCLC model. In this particular study, IL-18 had a greater effect than IL-12. IL-18 secretion by CAR-T cells may shift the immunosuppressive tumour environment to a pro-inflammatory state by increasing the activation of CD11b+Gr1– macrophages and MHC-II+CD11c+ dendritic cells at the tumour site and systemically, especially in SCLC, which contains a large number of macrophages. Data support the concept that antitumour immunity to large, established tumours can be improved in the long term by converting a chronic environment to an acute inflammatory environment [1,138].

### 9.2. Hypoxia-Responsive CAR-T Cells

Hypoxia is a feature of most solid tumours, and this feature may help distinguish CAR-T-cell tumours from normal tissues [139]. Tumour selectivity is a crucial factor because CAR-T-cell activity outside the tumour can lead to lethal toxicity. One preclinical study demonstrated selective expression of CAR-T cells targeting the entire ErbB within solid tumours, detecting hypoxia. These receptors are attractive targets for CAR therapy of solid tumours because they can be overexpressed in them. Despite the ubiquitous expression of ErbB receptors in healthy organs, this effect provides excellent antitumour efficacy without toxicity in healthy tissues. Although hypoxia is associated with immunosuppression, the killing capacity of T4-CAR-T cells is not negatively affected in hypoxic conditions. In the study, HypoxiCAR cells were generated and targeted to SKOV3 ovarian cancer cells. The tumour cells were also incubated with T4-CAR under normoxic and hypoxic conditions. HypoxiCAR cells demonstrated effective hypoxia-dependent killing of SKOV3 cells, comparable to that of T4-CAR cells. In contrast, in the in vivo study, HypoxiCAR was injected into HN3 tumour-bearing mice. After three days, no CAR molecules were detected on HypoxiCAR T cells collected from the blood, lungs, and liver, but CAR was expressed on the surface of the hypoxic TME. Tumour growth was effectively inhibited without systemic toxicity [140]. In yet another preclinical study, CAR-T cells were shown to be ineffective in the hypoxic environment of brain tumours. It was found that hypoxia in the tumour microenvironment leads to impaired mitochondrial ATP production in CAR-T cells, which impairs their effector functions. The authors proposed metabolic modifications that increase the resistance of these cells to hypoxia, improving their survival and efficacy in solid tumours, especially gliomas. This includes, among others, increasing the expression of factors that support cellular respiration and mitochondrial biogenesis, which would improve their survival and effector functions in hypoxic conditions, typical of gliomas [141]. However, these studies require validation in clinical trials. Further research is needed on the persistence, expression control, and potential impact of hypoxia on long-term T-cell effector function.

### 9.3. CAR-T Cells with Radiotherapy

Radiotherapy is a traditional form of cancer treatment. Combining CAR-T therapy with radiotherapy (RT) is a promising approach to overcome the limitations of the microenvironment in solid tumours. Radiotherapy can increase the expression of tumour antigens and MHC class I molecules on tumour cells, making them more susceptible to CAR-T-cell attacks. In addition, RT induces immunogenic cell death by releasing DAMPs and tumour antigens, which activate dendritic cells and enhance the immune response. Radiotherapy can also modulate the tumour microenvironment by increasing the expression of chemokines such as CXCL9 and CXCL10, which attract CAR-T cells to the tumour site. Additionally, RT can normalise tumour blood vessels by increasing the expression of adhesion molecules ICAM-1 and VCAM-1, which facilitates CAR-T-cell migration into the tumour. RT can also increase the expression of CAR-T target antigens, such as MUC-1 or CEA, on tumour cells, improving their recognition by CAR-T cells. However, RT can induce immunosuppression by increasing the number of Treg cells and secreting suppressive cytokines, which can inhibit CAR-T-cell activity. High doses of RT can damage CAR-T cells, so optimising the dose and timing of administration is crucial. Low doses of RT (<2–4 Gy) may be more beneficial, minimising toxicity and supporting the effect of immunotherapy. RT can also reduce tumour hypoxia, improving CAR-T-cell survival and function [142,143]. Caution should be exercised when designing combination therapy. In many patients, RT is given after CAR-T-cell administration. This poses an additional risk of CAR-T-cell apoptosis. However, it has been reported that memory CD8 cells are more resistant to apoptosis than naive T cells. In addition to the immunosuppression induced by increased numbers of Treg cells, TNF-beta should also be considered. DNA repair programs after RT treatment increase TNF-beta activity, resulting in immunosuppression. To overcome this barrier, CAR-T cells should be modified in such a way that they become resistant to TNF-beta-induced immunosuppression. Another aspect is the ATP released from irradiated tumour cells, which can be converted to adenosine, another barrier for CAR-T cells. Blocking adenosine signalling has been shown to induce a stronger T-cell response [144]. In the future, personalised approaches combining RT and CAR-T may increase the effectiveness of solid tumour treatment. Combining radiotherapy with CAR-T therapy presents a promising strategy for treating solid tumours, but further clinical and preclinical studies are needed to fully understand and optimise this therapeutic combination.

### 9.4. Nanotechnology in CAR-T Cells

Nanoparticles are a promising idea that could replace the virus in delivering CAR constructs to T cells, significantly reducing production costs [145]. Nanocarriers can be designed to deliver CAR-T cells directly to tumours, minimising the impact on healthy tissue. Additionally, encapsulating CAR-T cells in nanocarriers can protect them from the immune system, thereby improving the safety of the therapy. The goal is to enhance the durability and success of the treatment. Scientists from the University of Isfahan have also developed nanoparticles that can release immunomodulators and checkpoint inhibitors when combined with CAR-T cells [146]. Since overcoming physical and chemical barriers is a crucial element, nanogels or liposomes can facilitate the delivery of CAR-T cells to the tumour. Although brain tumours are a significant obstacle, scientists have created a tumour-targeted nanoparticle that degrades the extracellular matrix by converting light energy into local heat. Zhen Gu and colleagues proposed ICG-PLGA nanoparticles that, when activated by NIR light, loosen the ECM, which facilitates CAR-T activity. Zhu and his team created a photothermal PHCN nanozyme that, when heated, locally destroys the ECM in the tumour. Furthermore, in the case of brain tumours, nanoparticles can be modified to cross the blood–brain barrier via ligands that bind to receptors on endothelial cells. The nanoparticle can also function via ligands or antibodies that recognise tumours. This ability ensures that CAR-T cells are activated only in the tumour [147,148]. Adenosine inhibits the immune function of CD4+ and CD8+ T cells. The adenosine receptor on activated T cells causes adenosine to accumulate on the outside of the cell. This inhibits T-cell proliferation and INF-γ secretion. A promising strategy incorporates the use of nanotechnology to transport an adenosine receptor inhibitor to CAR-T cells in tumours. Several approaches have been suggested for using nanotechnology in CAR-T-cell therapy in solid tumours. However, the side effects of using nanotechnology in this area are still unknown, and further research is needed [148].

## 10. Discussion

### 10.1. Key Findings

This review highlights that CAR-T-cell therapy appears to face numerous unique challenges regarding the treatment of solid tumours, as opposed to its relatively successful use in haematological malignancies. These limitations include the immunosuppressive TME, antigen heterogeneity, toxicities, and limited tumour infiltration [1,12,81,116]. Overcoming these barriers may involve several strategies, such as hypoxia-responsive CAR constructs, cytokine-secreting “armoured” CAR-T cells, metabolic reprogramming, and targeted delivery approaches [1,3,20,104].

These strategies underline the importance of adapting CAR-T cells to the specific microenvironment of the tumour site, as well as combining different therapies [105,125]. These aim to ensure an improved and successful treatment that may not be achieved as efficiently as conventional CAR-T-cell therapy designs.

Notably, this review deepens novel approaches to CAR-T-cell therapy, including hypoxia-activated CARs, bispecific targeting strategies, and nanotechnology-assisted delivery [104,109,146]. Together, these studies show promise for future developments in the treatment of solid tumours using engineered CAR-T cells.

### 10.2. Preclinical Evidence

Numerous preclinical studies have demonstrated the efficacy and mechanisms of CAR-T-cell therapy for solid tumours. This includes an orthotopic model of human malignant pleural disease [75]. Additionally, mouse models with tumours such as glioma, ovarian cancer, and lung cancer were studied as well. These are used to test CAR-Ts targeting HER2, IL13Rα2, EGFR, and DLL3 [11,74,138]. Models focusing on hypoxia have further confirmed the potential of environment-responsive CAR-T designs [55]. Another preclinical study included an examination of glioblastoma models using CAR-T cells, which demonstrated local efficacy with limited systemic effects [107].

It is worth considering that there are several similarities and differences in the mechanisms, efficacy, and safety profiles of animal models and CAR-T-cell types used in the studies. Additionally, it is important to emphasise that preclinical models usually lack full fidelity to human immunology [149]. This limits the extent to which they can be reliably used in clinical settings. Both the FDA and EMA acknowledge that preclinical animal testing often has low predictive power for human efficacy and safety. These guidelines prioritise early clinical testing under strict monitoring over extensive non-human studies, recognising that CAR-T therapies’ unique immunobiological properties are best evaluated directly in human subjects [150]. The unnecessary use of animals in testing is also discouraged due to ethical reasons [151]. Overall, the limited translational gain from animal models does not fully justify routine animal testing.

Key similarities observed across studies include the following:Direct killing of tumour cells via granzyme and perforin secretion [1].

This mechanism is expected in solid tumour cell apoptosis and necrosis, as CAR-T cells are engineered to specifically target tumour antigens [114]. This finding underscores the universal potential of CAR-T-cell therapy in targeting and eliminating cancer cells.

Secondary immune activation via cytokine secretion (IFN-γ, IL-12, IL-18) [137].

This amplifies the overall therapeutic effect, supporting the potential of CAR-T-cell therapy to induce long-lasting antitumour immunity.

Enhanced tumour infiltration and persistence when combined with radiotherapy, oncolytic viruses, or nanotechnology [134,142,144,145,148].

The combination strategies suggest that CAR-T-cell therapy can be further optimised through multimodal approaches, as they help overcome barriers faced by CAR-T cells.

Key differences include the following:Route of administration: Local delivery generally produced better efficacy and lower systemic toxicity compared to intravenous infusion [102].

By ensuring a sufficient concentration of CAR-T cells at the tumour site, the potential of off-target adverse effects is greatly minimised [1]. This finding suggests that the method of CAR-T-cell delivery must be carefully considered for optimising therapeutic outcomes.

Cytokine profile: IL-18 often induced stronger acute inflammation than IL-12 [138].

This may result in more pronounced side effects such as CRS [81]. The variation in the cytokine profiles can be induced by the CAR construct and the tumour model being studied. This highlights the need for careful modulation of cytokine production to balance efficacy and minimise toxicity.

Hypoxia adaptation: CAR-T cells engineered for hypoxic environments showed greater tumour selectivity and safety [139].

CAR-T cells modified to recognise and respond to hypoxic conditions have a distinct advantage in selectively targeting these hard-to-reach regions while minimising damage to healthy tissue [141]. This advancement could lead to more effective and less toxic treatment for tumours with a hypoxic microenvironment.

Safety: Certain models reported neurotoxicity (particularly with CNS delivery) or off-target effects linked to antigen expression on healthy tissues [87].

These safety issues underscore the need for better tumour-specific targeting and caution when administering CAR-T cells to tumour sites. Variations in safety profiles across different tumour models further highlight the complexity of managing adverse effects in CAR-T-cell therapy.

The similarities observed across studies provide strong evidence that CAR-T-cell therapy holds significant potential for treating a broad range of solid tumours. However, the differences observed must be addressed before CAR-T-cell therapies can be successfully applied to treating those types of malignancies. The need to optimise the route of administration, adjust cytokine responses, engineer CAR-T cells for hypoxic environments, and manage safety concerns are critical issues that must be resolved. These findings also suggest that a one-size-fits-all approach may not be suitable for all tumour types, pushing the idea that CAR-T-cell therapies must be tailored to individual patients and tumour characteristics.

### 10.3. Clinical Evidence

The clinical trials provide insight into translating CAR-T-cell therapy in solid tumours from preclinical to clinical. Compared to preclinical studies, clinical studies remain more modest in their outcomes [117,119]. Early-phase trials targeting HER2-positive gastric cancer and EGFR-expressing non-small-cell lung cancer showed partial responses and temporary disease stabilisation, with progression-free survival typically limited to a few months [117,118]. These results highlight the challenges of tumour antigen heterogeneity, immune evasion, and treatment-related toxicity observed in patients but not fully replicated in animal models [116,119].

Clinical studies add real-world data on safety and attainability, confirming that most adverse reactions are manageable [117,118]. They also validate local delivery methods and expand on optimal dosing and manufacturing timelines [120,121,122]. However, their limited scale, short follow-up, and lack of patient diversity make it difficult to generalise the findings [119]. Diversifying the trials to include patients with different tumour types, genetic backgrounds, and treatment histories is essential to better understand the effectiveness of the therapy [13,152]. Furthermore, future trials should integrate combination therapies and next-generation CAR designs that address the hostile TME, as preclinical studies suggest these modifications may unlock greater clinical benefit [137,138,139,140,141].

### 10.4. Strengths and Limitations of Existing Research

CAR-T-cell therapy offers a treatment which, if managed appropriately, does not require extensive hospitalisation and has a quick initial response rate. Notably, there is also potential for long-term remission, with toxicities that are often short-term and reversible in comparison to those caused by traditional treatment [83,88]. The therapy brings novel engineering concepts into oncology, paving the way for innovative treatment models.

Despite promising results, many aspects of CAR-T-cell therapy in solid tumours remain controversial or understudied. Its toxicities, while controllable, still pose a threat to the patient’s life. Facilities providing the treatment have to be properly equipped, with staff trained to specifically recognise symptoms of CRS and ICAN. This poses additional costs for hospitals and can subsequently limit the number of clinics providing CAR-T therapy and its accessibility.

Most studies are short-term, with heterogeneous patient populations and limited sample sizes [97]. Many preclinical models fail to fully replicate the human TME and immune responses, limiting translational value [13]. Optimal routes of administration remain uncertain. While local delivery is promising, it may not be feasible for multifocal disease [114]. Study designs vary widely in CAR constructs, targets, and treatment protocols, complicating cross-trial comparisons [1,3,7,11]. There is a lack of validated biomarkers to predict which patients will respond best [1,11,13]. Moreover, the number of patients included in and completing these studies is underwhelming. This limits the statistical power of the results and prevents an assessment of effectiveness in a broader population [7,102].

### 10.5. Implications and Future Directions

Enhancement in the therapeutic potential and safety of CAR-T-cell therapy requires several factors that are promising for future exploration.

Reprogramming the immunosuppressive TME into an acute inflammatory state, by secreting IL-12 and IL-18, has shown potential to activate macrophages and dendritic cells and increase cell infiltration and survival. Although the results are based on preclinical data, this topic is worth further exploration.

A combination of CAR-T cells and radiotherapy to enhance antigen presentation and modulate the TME is a promising future direction; however, it requires more in-depth studies aiming to decrease radiation-induced apoptosis and immunosuppression.

Nanotechnology in CAR-T-cell therapy is another promising future direction due to its reduced impact on healthy tissues, the possibility of crossing the blood–brain barrier, and the selective recognition of tumours. Additionally, nanotechnology could reduce the production cost, making the therapy more accessible to a wider range of patients. Despite the positive opportunities, nanotechnology in this area requires further research.

## 11. Conclusions

CAR-T-cell therapy has emerged as a revolutionary treatment, primarily in haematological cancers, by providing a high initial response and potentially long-term remission. However, there are several significant obstacles to safe and effective implementation, including cytokine release syndrome and immune effector cell-associated neurotoxicity syndrome. Although the side effects are mostly acute and reversible, awareness of their occurrence and the ability to counteract them may significantly improve treatment safety. Usage of CAR-T-cell therapy in solid tumours is still challenging, but there are possibilities to increase the effectiveness and quality of this treatment. One of the most challenging factors is the immunosuppressive tumour microenvironment, which leads to decreased efficiency. However, a promising approach yet to be proven clinically is based on engineering CAR-T cells to produce IL-12 and IL-18 and converting the environment to an acute inflammatory state, enhancing T-cell activation and tumour control, especially in macrophage-rich tumours like SCLC. Another approach is strictly connected to hypoxia, which is a common feature among solid tumours. Using CAR-T cells reactive in a hypoxic environment reduces the possibility of off-tumour toxicity and makes maintaining effective killing possible. However, a hypoxic environment can cause impairment of mitochondrial metabolism, according to present studies, mostly in gliomas, which can be countered by metabolic modifications to maintain survivability and efficiency. Radiotherapy combined with CAR-T-cell therapy is promising, mostly because of increased antigen presentation, better infiltration, and reduced hypoxic environment. CAR-T-cell administration is typically carried out before radiotherapy, which may potentially increase the risk of apoptosis. However, using lower doses of radiotherapy and genetic modifications leads to positive results. This innovative approach may significantly improve CAR-T-cell therapy outcomes in solid tumours.

## Figures and Tables

**Figure 1 biology-14-01035-f001:**
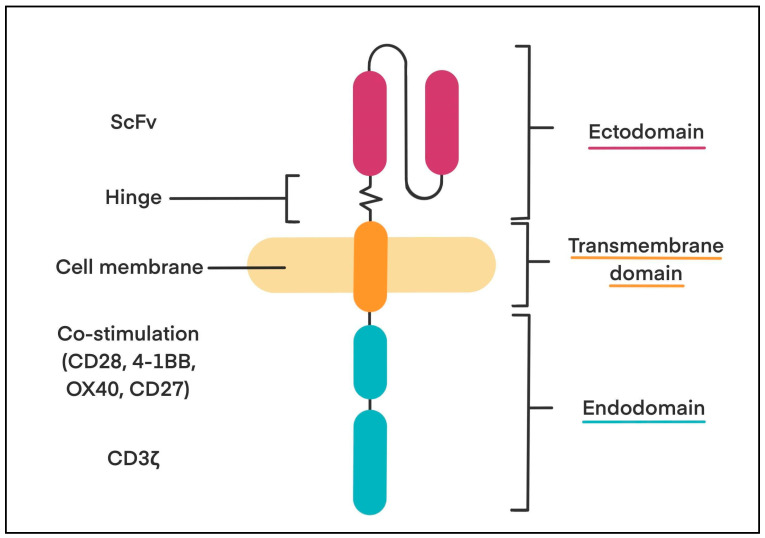
Schematic representation of CAR structure. ScFV—single-chain variable fragment; CD—cluster of differentiation.

**Figure 2 biology-14-01035-f002:**
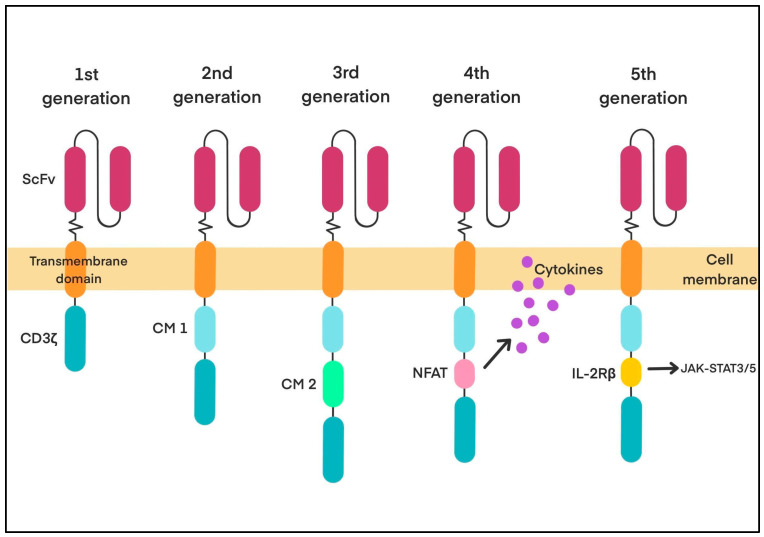
Schematic representation of CAR-T-cell generations. ScFV—single-chain variable fragment; CD—cluster of differentiation; CMs—co-stimulatory molecules; NFAT—nuclear factor of activated T cells; JAK—Janus kinase; STAT—signal transducer and activator of transcription.

**Figure 3 biology-14-01035-f003:**
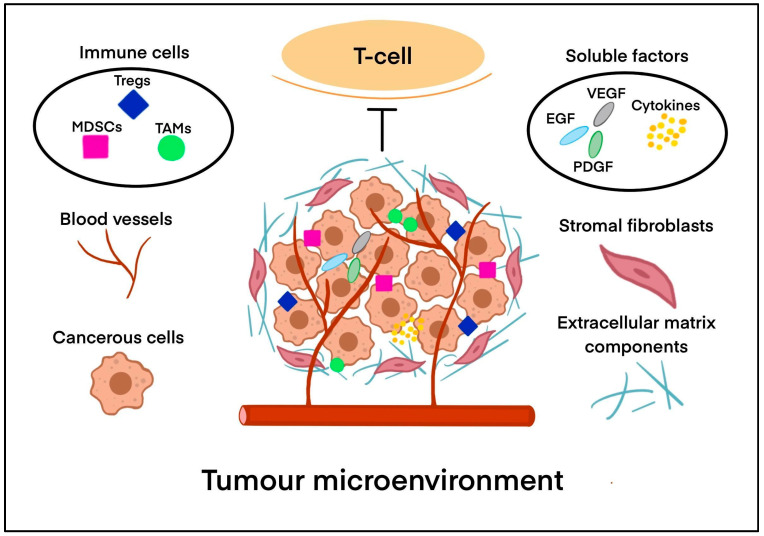
Schematic representation of the immunosuppressive tumour microenvironment. MDSCs—myeloid-derived suppressor cells; TAMs—tumour-associated macrophages; Tregs—regulatory T cells.

**Table 1 biology-14-01035-t001:** Common toxicities of cancer therapies.

Toxicity Type	Chemotherapy	Radiotherapy	CAR-T Therapy
Gastrointestinal toxicity	Nausea (96.5%) [42], emesis (acute, delayed, anticipatory), malnutrition, acid–base imbalance [34,35], diarrhoea, and mucositis [36].	Nausea, emesis, and diarrhoea [43].	Nausea [47,48].
Cardiotoxicity	Chronic heart failure, hypertension, arrhythmias, and cardiomyopathy [37].	Valvular heart disease (60%) [45], myocarditis, and fibrosis [43,45,46].	In severe cases of CRS [47,48].
Nephrotoxicity	Acute kidney injury and chronic renal failure risk [38,40].	Radiation-induced kidney injury [74].	Low incidence of acute kidney injury [75].
Haematological toxicity	Myelosuppression, anaemia, and thrombocytopenia [41].	Marrow suppression (depending on site) [43].	Cytopenias * and CRS-related systemic inflammation [47].
Neurological toxicity	Polyneuropathies [41].	Late cognitive impairment [43].	ICANS: confusion, aphasia, and seizures [49].
Immunologic toxicity	Hypersensitivity reactions [33].	Dermatitis and pneumonitis from local immune effects [43].	CRS (50–90%) [47,48].
Other	Alopecia (98.6%), anorexia (97.9%), and dysgeusia (97.2%) [42].	Lymphoedema (breast cancer radiotherapy: 20.4%; overall: 42.2%) [44], alopecia, and dermatitis [43].	Fever [47,48].
Reversibility	Chronic or irreversible (e.g., cardiotoxicity, neuropathy) [37,40].	Acute symptoms (reversible); late fibrosis and cardiac effects may persist [43].	Mostly reversible; long-term organ damage risk [49].

* Early cytopenias are expected after CAR-T-cell treatment; however, there are occurrences of prolonged cytopenias of yet unknown pathophysiology [76]. ICANS—immune effector cell-associated neurotoxicity syndrome.

**Table 2 biology-14-01035-t002:** CAR-T-cell therapy in solid tumours—challenges and clinical advances.

Challenge	Findings	CAR-T Strategy	Clinical/Preclinical Results	Risks and Limitations	References
Antigen heterogeneity	Solid tumours lack unique, uniformly expressed antigens; shared expression with healthy tissues increases toxicity risks	—	—	High risk of off-tumour toxicity; antigen escape	[116]
HER2-targeted CAR-T in gastric cancer (preclinical)	HER2 identified as a viable target; CAR-Ts show selective cytotoxicity	HER2-specific CAR-T cell	Eliminated HER2+ patient-derived cells; strong activity in xenograft models; MHC-independent apoptosis induction	Limited to HER2+ cancers; need for careful patient selection	[117]
HER2-targeted CAR-T in gastric cancer (clinical)	Evaluation of safety and efficacy in HER2+ advanced cancer patients	HER2-specific CAR-T cell	1 partial response (4.5 mo); 5 stable disease cases (median PFS: 4.8 mo)	1 patient developed severe GI bleeding	[117]
EGFR-targeted CAR-T in NSCLC (phase I trial 1)	Investigated efficacy in EGFR-expressing NSCLC	EGFR-specific CAR-T cell	4 partial responses (2–4 mo); 8 stable disease; manageable grade 3 AEs; median PFS: 3 mo; OS: 4.9 mo	Short duration of response; toxicity manageable but present	[118]
EGFR-targeted CAR-T in NSCLC (phase I trial 2)	Follow-up in refractory/relapsed NSCLC	EGFR-specific CAR-T cell	1 durable partial response (>13 mo); 6 stable disease; median PFS: 7.13 mo; OS: 15.63 mo; all AEs were manageable	Small sample size; early-phase trial	[118]
General clinical landscape	Solid tumours still pose major hurdles due to TME and immune evasion	Various under investigation	Some promising early results in HER2+ and EGFR+ cancers	No major breakthroughs yet; still investigational	[119]

CAR-T cell—chimeric antigen receptor T cell; EGFR—epidermal growth factor receptor; HER2—human epidermal growth factor receptor 2; NSCLC—non-small-cell lung cancer; TME—tumour microenvironment.

**Table 3 biology-14-01035-t003:** Combination strategies to overcome immunosuppression in CAR-T therapy for solid tumours.

Approach	Mechanism	Key Outcomes	Phase of Study	References
Checkpoint Inhibitors (e.g., anti-PD-1/PD-L1)	Restore CAR-T function by reversing T-cell exhaustion and improving persistence	Enhanced CAR-T proliferation, persistence, and antitumour response in early studies	Clinical trial	[107]
Oncolytic Viruses	Induce tumour cell lysis, release tumour antigens, promote inflammation, and recruit T cells into cold tumours	OV + CAR-T shows superior tumour control and survival in solid tumour models	Preclinical model, early clinical trials	[123,134]
Low-dose Chemotherapy/Radiotherapy	Lymphodepletion reduces suppressive immune cells (Tregs, MDSCs) and prepares niche for CAR-T expansion	Improved CAR-T engraftment and synergy in trials combining radiotherapy with EGFRvIII-targeted CAR-T cell	Clinical trial	[125,126]
Metabolic Checkpoint Inhibitors	Inhibit IDO and A2A receptors to reduce kynurenine and adenosine-mediated T-cell suppression	Enhanced CAR-T function under metabolic stress; adenosine deaminase overexpression improves antitumour activity	Preclinical model clinical trials	[127,128]
RNA/mRNA Vaccines	Boost CAR-T and endogenous T-cell immunity; promote epitope spreading; overcome antigen escape	Amplifying RNA vaccines improve CAR-T expansion and responses; mRNA boosters expand polyclonal immunity	Clinical trial	[129,130,131]
Synthetic Notch/Logic-Gated CARs	Conditional activation improves specificity and reduces off-tumour toxicity	Enhanced discrimination between tumour and healthy tissue; precise response control	Preclinical models, early clinical trial	[133]
Kinase Inhibitors (e.g., Dasatinib)	Reversible CAR-T inhibition allows functional recovery and prevents exhaustion	Dasatinib temporarily rests CAR-T cells and improves their long-term function	Preclinical model	[135]

CAR-T cell—chimeric antigen receptor T cell; EGFRvIII—epidermal growth factor receptor variant III; EGFR—epidermal growth factor receptor; MDSCs—myeloid-derived suppressor cells; OV—oncolytic virus; PD-1—programmed cell death protein 1; PD-L1—programmed death ligand 1; Tregs—regulatory T cells.

## Data Availability

Not applicable.

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
