# Peer review of "Can We Use CAR-T Cells to Overcome Immunosuppression in Solid Tumours?"

_biology, 2025, doi:10.3390/biology14081035_

Round 1

Reviewer 1 Report

Comments and Suggestions for Authors

The manuscript of Gwadera J et al is an extensive, (text body of 25 pages) long review entitled to focus into the question whether use of CAR-T cells can overcome the immunosuppression in solid tumour. The authors’ describe commendably the existing basic immunological mechanisms of the three main entities. In contrast, presentation of preclinical and clinical studies can be improved. Also, the value of the review would increase by sharpening the authors’ standpoints of reached knowledge of preclinical data and climical implementations. Similarly, the review would benefit of clear opinion of gained experience as well as suggestions for future research direction. The abstract is written more attractive way than the text body offers information on and the conclusion is too generall.

Major comments

  1. Since CAR-T cells as gene therapy products, thus presenting one of the latest technological advances in medical research and are classified as advanced therapy medicinal products based on both European (EMA/EU) and American (FAD) legislation, this information could be included early in the introduction subsection of chapter 1. Here also could be explained that the donated autologous or allogeneic lymphocytes constitute starting material, i.e. cell population for engineering T lymfocytes with CARs to gain the drug.
  2. The text body is long and repeats basic information in subsections. Some sentences introdusing and/or concluding subsections are proposed to be deleted, especially when not delivering specific statements of new results or advantages. This will improve the reading and delivering the most important message.

One proposal to shorten the text body is to reorganize the chapters to avoid lot of repeating text. Also, the reorganisation would bring more clear and logical order of the presentation.

2a) The long text body in chapters 1 and 2 is proposed to be shortened to only focus in those elements that are crusical to the topic of the review. The text is mainly basic immunology and could refer to an newly updated edition of a text book. The last subsection on clinical experience in the chapter 1 is proposed to be largly shortened or totally deleted since the data is presented later, even partly repeated twice, both in text and tables.

2b) Since chapters 3 and 4 are presenting clinical experience, they may better suit to be presented later in the text body, i.e. after presenting approaches to overcome problems with immunosuppressive microenvironment of solid tumours and manipulation/engineering of CAR-T cells (chapters 5,6). This would give a smooth brigde to continue with chapter 7.

2c) Chapters 4 to 6 are repeating information of faced difficulties to gain success in implementing CAR-T cell therapy in solid tumour treatment. Please, evaluate the importance of the chapter 4, since the most of data is identified in later chapters and/or could be added into reorganized text body. The chapter 5 is the most informative and comprehensive one.

2d) Please, combine the data of modulling the tumour microenvironment (TME) collecting approaches studied concerning tumour-associated antigens, immunosuppressive cells and mediating substances, physical/chemical factors (pH, hypoxia, deprivation of other essential nutrients for cell survival in TME, etc). Also clearly point out when data is preclinical and clinical, respectively, as well as give critical evaluation of the value of used preclinical models. Immunological data from in vitro studies and preclinical studies on small animals face major difficulties for any conclusions concerning expected success in patients.

2e) The presentation of additional approaches to improve the outcome of CAR T cell therapy, eg. delivering the cells, properties of the manufactured end product concerning the proportions of CD4+ and CD8+ as well as possible other cell components is lacking authors critical evaluation of collected data to guide future development. What other cell rations than 1:1 using CD4+ and CD8+ have been studied and is 1:1 ratio the naturally existing one in TME or why would this ratio be one for further development? Would delivering CD4+ cells separately from CD8+ cells using a predefined time intervall and order give further success? Would these interventions, being a potential prolongation of manufacturing time and thus a negative element overcome benefits in patient care and overall results? Please, improve the message of these aspects.

3) The figures and tables are wellcomed, and should be referred to in each applicable location in the text body. Observe that tables are to be able to be studied without checking the text body. Thus, please add used abbreviations, number of patients included clinical trials, whether presented study was preclinical or clinical, etc. Table may have notes below the lay-out form.

4) The text body in chapter 2 (and even in later ones) would benefit from added information whether the statements of various findings and experiences are based on preclinical and/or clinical data, and whether preclinical data are comfirmed in clinical studies including number of patients studied and thus demonstrating any credible value of the preclinical experience. The authors’ evaluation and critical standpoint would be valuable to guide further research and development of this treatment modality.

5) Presentation of various generations of CAR-T cells is lacking information of the 5th generation. Please, complement on page 3/25.

6) Presentation of toxicity of chemo- and radiotherapy (sections 3.1 and 3.2) is wellknown information. Thus, the text is proposed to be shortened and references made to the table 1.

7) Concerning mild CRS (grade 1) as compared to more severe ones, please, check and revise in the section 3.3, lines 5-7. The first sign of CRS is usually fever. Mild CRS (Grade 1) can be managed conservatively (antipyretics, fluid). Hypotension is already a sign of higher grade toxicity. All higher grade symptoms require intensive monitoring and interventions. Starting from Grade 2, an admission to an Intensive Care Unit should be considered. The last sentence could preferably be complemented with information of requirement of CAR-T cell treatments to be performed at qualified clinics with experience of experimental cell and gene therapies (gained for example via previous participation in clinical trials of these high-risk therapies).

8) The title of section 3.4 uses a comparative form, ’Shorter’, and the text gives an impression of benefit of using CAR-T cell therapy but doesn’t clearly inform what a shorter treatment duration is compared with. No data is given about treatment time lines of chemo- and radiation therapies that obviously are the therapies the authors have aimed to discuss results of time lines when CAR T cells are used. In addition, various health care systems and organisations should be added elements to the list of divergent data collected in various clinical trials on patient care duration at a hospital. Preferably the authors will also add here the requirement to treat patients at a qualified hospital clinic as facilities to stress the risks of a new therapy form.

9) The presentation of chemo- and radiation therapy would benefit of pointing out the aim of these therapies when used in combination of CAR-T cells. Specify when the data describing these therapies are used with tumour load decreasing goal and/or to avoid CAR-T cell rejection and/or to prepare the tumour environment to be less immunosuppressive or any additional purpose. Depending of the aim, the order of given combination therapy as well as its timing are awaited to be discussed and evaluated by concluding the authors’ opinion.

10) The presentation of clinical studies using CAR-T cells would benefit of bringing up known differences of for example melanoma and renal cell cancer as compared with colorectal, lung, ovarian cancer. The former are not treatable using chemotherapy but immunological active agencies. How does this influence the focus in choosing which solid tumour types would be expected to be successfully treated using CAR-T cells?

11) The abstract needs to be revised to mirror the information in the text body and summared conclusions and recommendations. The present abstract form is promising more than the manuscript offers.

12) The manuscript is missing a separate chapter ’Discussion’ with critical evaluation of published and presented preclinical and clinical experience as well as constructive recommendations for future research.

13) The conclusions need to collect the focus areas such as need to decrease the CAR-T cells toxicity, to improve applications of knowledge of TME and specific behavior of various tumour types, to improve understanding of mechanisms of various therapy entities and combinations of them, etc based on authors’ comprehensive analysis of the existing literature including identifying current gaps and problems. The main conclusion is expected to be critical and constructive and provide recommendations for future research.

Minor comments

    1. The abbreviations need to be checked concerning their position to be used for the first time. For example, on page 2/25, line 1, ’tumour-associated antigen’ can already be noted by (TAA) and thus, always referred to later on. Please, check also for example CAR, TMD, cTCR, MHC, CD134, OX-40, 2G, etc. Observe that generally one doesn’t use an abbreviation in the beginning of a sentence.

  1. Please, add more often in the text body references to figures and table.
  2. Please, check the reference number 109 included on page 4/25, line 11: ’… immune response agains the tumour109. False numbering, writing??
  3. Please, pay attention to some statements’ power and/or quality. For example, the statement of nausea and emesis to be the most feared side effect of chemotherapy can be questionable as compared to risk and severe consequences of organ toxicity.
  4. Please, add reference to the agreement (consensus fonference or comparable forum) of grading to 1-4 CRS toxicity of CAR-T cells (section 3.3).
  5. Please, check the data of response rate of tocillizumab used in the marketing approval of the drug (EMA and/or FAD) to more correctly inform of the valuable benefit and add a corecponding reference to data of response rate.
  6. Please, check the writing in the subsection 2.1.3, lines 3-6: the two sentences are repeating almost similar information (can be merged).
  7. Please, check the English writing in section 3.3, last subsection, line 2: ’They are more …’, since when using comparative (better than/more than) one expects information of what the statement is comparing with each other. Also, on line 3, there might be a false ’-’ sign? Would the sign ’.’ sign improve the reading?
  8. Concerning the table 1, the nausea is presented twice as a toxicity of chemotheapy confusing the understanding of the information the authors are aiming to give. Please, specify. Futher more, cytopenia is given as a haematological toxicity of CAR-T cell therapy. The question is whether this is thrue per se for CAR T cells or more a side effect of preparative therapy to be given prior to CAR T cells and/or a combination of both. Please, specify for example by giving additional explanation in notes to the table. Concerning immunologic toxicities of CAR T cells, ’cytokine release syndrome’ can be given as an abbreviation ’CRS’ as already used earlier in the same column among haematoligcal toxicities.
  9. Please, give the correct reference to the table 2 on the last line of the chapter 4 (no writing ’… in Table 1’.
  10. Please, add information of whether data in the Table 3 is from a preclinical and/or clinical study. This will improve reading since a reader should not need to check references in the the text body to study a table.
  11. Please, re-evaluate if the last row in the tables is necessary (eg. presented as ’Summary outlook’ or ’Overall Strategies’). What is the additional information as compared to the other rows presenting the detailed data?
  12. It seems that in chapters 7-10 several references that already are presented in earlier chapters presenting the same topics are missing. Please, add additional references.
  13. It seems that there is a writing misstake in the beginning of section 10.4, line 1-2 when writing ’… in delivering CAR cells to T cells,…’? CARs are not cells but the specific receptors engineered into T cells. Please, check the writing.
Comments on the Quality of English Language

Please, see among 'Minor comments'.

Author Response

Dear Reviewer 1,

Thank You very much for Your time and effort, as well as constructive comments concerning our manuscript. We have studied Your comments carefully, had team discussions, sometimes difficult, and made few significant corrections and improvements, which we hope to meet with Your approval. Thanks to this work being done, we were also able to expand our knowledge even more - and enhance our future projects.

We decided to highlight all added or changed paragraphs to facilitate the process of reviewing and save Your time. Additionally, we’ve tried to explain our point of view in several points below.

We really hope these modifications can meet with Your approval.

Thank You very much for Your time and effort. We know that nowadays time is scarce. 

 Yours Sincerely,

Authors

  1. Presentation of preclinical and clinical studies can be improved.

Thank you for your review. We have expanded and clarified the presentation of preclinical and clinical studies accordingly.

  1. The value of the review would increase by sharpening the authors’ standpoints of reached knowledge of preclinical data and clinical implementations.

We agree, thank you. This has been included.

  1. Similarly, the review would benefit from a clear opinion of gained experience as well as suggestions for future research direction.

Thank you, we included this in a new chapter - ‘Discussion’, where we included your suggestions.

  1. Since CAR-T cells as gene therapy products, thus presenting one of the latest technological advances in medical research and are classified as advanced therapy medicinal products based on both European (EMA/EU) and American (FAD) legislation, this information could be included early in the introduction subsection of chapter 1.

Thank you, this has been included.

  1. Here also could be explained that the donated autologous or allogeneic lymphocytes constitute starting material, i.e. cell population for engineering T lymphocytes with CARs to gain the drug.

This has been changed. Thank you.

  1. The text body is long and repeats basic information in subsections. Some sentences introducing and/or concluding subsections are proposed to be deleted, especially when not delivering specific statements of new results or advantages. This will improve the reading and delivering the most important message. The long text body in chapters 1 and 2 is proposed to be shortened to only focus in those elements that are crucial to the topic of the review. The text is mainly basic immunology and could refer to an newly updated edition of a text book. The last subsection on clinical experience in the chapter 1 is proposed to be largely shortened or totally deleted since the data is presented later, even partly repeated twice, both in text and tables.

Thank you. We have implemented the suggested changes. However, we believe that the extended content in Chapter 1 and 2 is important for a better understanding of CAR-T therapy overview and the tumour microenvironment, which is crucial for recognising the mechanisms and challenges of CAR-T cell therapy. The summary tables are included as a visual aid to organise the key information more effectively and enhance clarity.

  1. Since chapters 3 and 4 are presenting clinical experience, they may better suit to be presented later in the text body, i.e. after presenting approaches to overcome problems with immunosuppressive microenvironment of solid tumours and manipulation/engineering of CAR-T cells (chapters 5,6). This would give a smooth bridge to continue with chapter 7.

This has been done, thank you for your comment.

  1. Chapters 4 to 6 are repeating information of faced difficulties to gain success in implementing CAR-T cell therapy in solid tumour treatment. Please, evaluate the importance of the chapter 4, since the most of data is identified in later chapters and/or could be added into reorganized text body. The chapter 5 is the most informative and comprehensive one.

Thank you for your valuable feedback. We have reorganized the manuscript accordingly, integrating the relevant data from Chapter 4 into other sections to improve clarity and coherence. We appreciate your insightful comments, which have helped enhance the overall structure of the paper.

  1. Please, combine the data of modelling the tumour microenvironment (TME) collecting approaches studied concerning tumour-associated antigens, immunosuppressive cells and mediating substances, physical/chemical factors (pH, hypoxia, deprivation of other essential nutrients for cell survival in TME, etc). Also clearly point out when data is preclinical and clinical, respectively, as well as give critical evaluation of the value of used preclinical models. Immunological data from in vitro studies and preclinical studies on small animals face major difficulties for any conclusions concerning expected success in patients.

Thank you for your suggestion. We have incorporated the data on tumour microenvironment modelling by consolidating information on tumour-associated antigens, immunosuppressive cells and mediators, as well as physical and chemical factors such as pH, hypoxia, and nutrient deprivation. Additionally, we have included data from in vitro studies in another chapter. While we recognize that the tumour microenvironment is an important and fascinating topic, we believe it warrants separate, dedicated reviews. In this manuscript, we chose to maintain a focused perspective on CAR-T cell therapy.

  1. The presentation of additional approaches to improve the outcome of CAR T cell therapy, eg. delivering the cells, properties of the manufactured end product concerning the proportions of CD4+ and CD8+ as well as possible other cell components is lacking authors critical evaluation of collected data to guide future development. What other cell rations than 1:1 using CD4+ and CD8+ have been studied and is 1:1 ratio the naturally existing one in TME or why would this ratio be one for further development? Would delivering CD4+ cells separately from CD8+ cells using a predefined time intervall and order give further success? Would these interventions, being a potential prolongation of manufacturing time and thus a negative element overcome benefits in patient care and overall results? Please, improve the message of these aspects.

Thank you for your comment and the explanation. We truly appreciate your feedback, as it highlights important details that will help us improving the manuscript. The section has been revised accordingly, and the relevant information regarding CD4+ and CD8+ cell ratios, delivery strategies, and their implications on manufacturing and therapeutic outcomes has been added.

  1. The figures and tables are wellcomed, and should be referred to in each applicable location in the text body. Observe that tables are to be able to be studied without checking the text body. Thus, please add used abbreviations, number of patients included clinical trials, whether presented study was preclinical or clinical, etc. Table may have notes below the lay-out form.

Thank you. We have added the requested information to the tables, including explanations of abbreviations and clear indications of whether the studies are preclinical or clinical. Additionally, we have expanded the notes below the tables to ensure they can be understood independently of the main text.

  1. The text body in chapter 2 (and even in later ones) would benefit from added information whether the statements of various findings and experiences are based on preclinical and/or clinical data, and whether preclinical data are confirmed in clinical studies including number of patients studied and thus demonstrating any credible value of the preclinical experience. The authors’ evaluation and critical standpoint would be valuable to guide further research and development of this treatment modality.

Thank you for your comment. We have addressed this point in the subsequent chapters by clearly distinguishing between preclinical and clinical data. However, taking into account suggestions from other reviewers, we decided not to expand this aspect in Chapter 2 in order to maintain a focused discussion on the mechanisms within the tumour microenvironment.

  1. Presentation of various generations of CAR-T cells is lacking information of the 5th generation. Please, complement on page 3/25.

This has been done, thank you for your comment.

  1. Presentation of toxicity of chemo- and radiotherapy (sections 3.1 and 3.2) is wellknown information. Thus, the text is proposed to be shortened and references made to the table 1.

This has been done, thank you for your comment.

  1. Concerning mild CRS (grade 1) as compared to more severe ones, please, check and revise in the section 3.3, lines 5-7. The first sign of CRS is usually fever. Mild CRS (Grade 1) can be managed conservatively (antipyretics, fluid). Hypotension is already a sign of higher grade toxicity. All higher grade symptoms require intensive monitoring and interventions. Starting from Grade 2, an admission to an Intensive Care Unit should be considered. The last sentence could preferably be complemented with information of requirement of CAR-T cell treatments to be performed at qualified clinics with experience of experimental cell and gene therapies (gained for example via previous participation in clinical trials of these high-risk therapies).

Thank you for pointing this out, I have reevaluated my resources and improved this section accordingly.

  1. The title of section 3.4 uses a comparative form, ’Shorter’, and the text gives an impression of benefit of using CAR-T cell therapy but doesn’t clearly inform what a shorter treatment duration is compared with. No data is given about treatment time lines of chemo- and radiation therapies that obviously are the therapies the authors have aimed to discuss results of time lines when CAR T cells are used. In addition, various health care systems and organisations should be added elements to the list of divergent data collected in various clinical trials on patient care duration at a hospital. Preferably the authors will also add here the requirement to treat patients at a qualified hospital clinic as facilities to stress the risks of a new therapy form.

Thank you very much for your insightful comments. We have introduced minor revisions in section 3.4, though we chose to keep the discussion focused. We appreciate your suggestion regarding including treatment timelines for chemotherapy and radiation therapies as well as the influence of different healthcare systems and the importance of qualified hospital settings for patient safety. These are valuable points that can be further explored in future work.

We are deeply grateful for your time and valuable hints. For many of us, this is our very first revision, and we are committed to continuous improvement and learning in the future.

  1.  The presentation of chemo- and radiation therapy would benefit of pointing out the aim of these therapies when used in combination of CAR-T cells. Specify when the data describing these therapies are used with tumour load decreasing goal and/or to avoid CAR-T cell rejection and/or to prepare the tumour environment to be less immunosuppressive or any additional purpose. Depending of the aim, the order of given combination therapy as well as its timing are awaited to be discussed and evaluated by concluding the authors’ opinion.

Thank you for your valuable suggestion. We have made minor adjustments to the text; however, after careful consideration, we have decided to retain the current structure and presentation of this section. We believe the existing format sufficiently addresses the role of chemo- and radiation therapies in combination with CAR-T cells within the scope of our review.

  1.  The presentation of clinical studies using CAR-T cells would benefit of bringing up known differences of for example melanoma and renal cell cancer as compared with colorectal, lung, ovarian cancer. The former are not treatable using chemotherapy but immunological active agencies. How does this influence the focus in choosing which solid tumour types would be expected to be successfully treated using CAR-T cells?

Thank you very much for your insightful comment. We appreciate the suggestion to elaborate on the differences between tumor types such as melanoma and renal cell carcinoma compared to colorectal, lung, and ovarian cancers. After careful consideration, we have decided to maintain the current scope and focus of our manuscript without expanding extensively on this topic. Our rationale is that the primary aim of the review is to provide a broad overview of CAR-T cell therapy applications across various solid tumors, while detailed tumor-specific therapeutic nuances may warrant a dedicated, in-depth analysis beyond the scope of the present work.

  1. The abstract needs to be revised to mirror the information in the text body and summated conclusions and recommendations. The present abstract form is promising more than the manuscript offers

Thank you for this remark, indeed, following your advice we’ve decided to improve our manuscript rather than solely the abstract. Minor changes have been made, certainly, however major changes can be seen in our paper. We’re deeply grateful for your time and valuable hints - for many of us this is the very first revision paper and we’ll do our best in the upcoming future to improve and learn.

  1. The manuscript is missing a separate chapter ’Discussion’ with critical evaluation of published and presented preclinical and clinical experience as well as constructive recommendations for future research.

Thank you for your valuable comment. We have addressed this by adding a separate ‘Discussion’ chapter that critically evaluates the published and presented preclinical and clinical data, along with constructive recommendations for future research.

  1.  The conclusions need to collect the focus areas such as need to decrease the CAR-T cells toxicity, to improve applications of knowledge of TME and specific behavior of various tumour types, to improve understanding of mechanisms of various therapy entities and combinations of them, etc based on authors’ comprehensive analysis of the existing literature including identifying current gaps and problems. The main conclusion is expected to be critical and constructive and provide recommendations for future research.

Thank you very much for your comment and for taking the time to review our manuscript. We have incorporated additional information in the ‘Conclusions’ section addressing key focus areas, including the need to reduce CAR-T cell toxicity. We have highlighted current CAR-T cell therapy challenges such as cytokine release syndrome and immune effector cell-associated neurotoxicity syndrome. Additionally, we discussed promising strategies, which may improve treatment efficacy in the future.

  1. The abbreviations need to be checked concerning their position to be used for the first time. For example, on page 2/25, line 1, ’tumour-associated antigen’ can already be noted by (TAA) and thus, always referred to later on. Please, check also for example CAR, TMD, cTCR, MHC, CD134, OX-40, 2G, etc. Observe that generally one doesn’t use an abbreviation in the beginning of a sentence.

Thank you, this has been done.

  1. Please, add more often in the text body references to figures and table.
    Thank you, the references have been added.
  2. Please, check the reference number 109 included on page 4/25, line 11: ’… immune response agains the tumour109. False numbering, writing??

Thank you for pointing this error out, we have corrected it.

  1. Please, pay attention to some statements’ power and/or quality. For example, the statement of nausea and emesis to be the most feared side effect of chemotherapy can be questionable as compared to risk and severe consequences of organ toxicity.

The wording has been changed, thank you for your comment.

  1. Please, add reference to the agreement (consensus conference or comparable forum) of grading to 1-4 CRS toxicity of CAR-T cells (section 3.3).

The reference has been added, thank you for your comment.

  1. Please, check the data of response rate of tocillizumab used in the marketing approval of the drug (EMA and/or FAD) to more correctly inform of the valuable benefit and add a corresponding reference to data of response rate.

This has been done, thank you for your comment.

  1. Please, check the writing in the subsection 2.1.3, lines 3-6: the two sentences are repeating almost similar information (can be merged).

Thank you for your comment. We fully agree — the sentences were repetitive, and we have revised them accordingly.

  1. Please, check the English writing in section 3.3, last subsection, line 2: ’They are more …’, since when using comparative (better than/more than) one expects information of what the statement is comparing with each other. Also, on line 3, there might be a false ’-’ sign? Would the sign ’.’ sign improve the reading?

This has been done, thank you for your comment.

  1. Concerning the table 1, the nausea is presented twice as a toxicity of chemotheapy confusing the understanding of the information the authors are aiming to give. Please, specify. Futher more, cytopenia is given as a haematological toxicity of CAR-T cell therapy. The question is whether this is thrue per se for CAR T cells or more a side effect of preparative therapy to be given prior to CAR T cells and/or a combination of both. Please, specify for example by giving additional explanation in notes to the table. Concerning immunologic toxicities of CAR T cells, ’cytokine release syndrome’ can be given as an abbreviation ’CRS’ as already used earlier in the same column among haematoligcal toxicities.

Thank you for your comment. We have resolved the issue regarding the repeated mention of nausea in Table 1 to avoid confusion. Additionally, we included a clarifying note below the table addressing the origin of cytopenia in the context of CAR-T cell therapy. As suggested, we have also used the abbreviation ‘CRS’ abbreviation.

  1. Please, give the correct reference to the table 2 on the last line of the chapter 4 (no writing ’… in Table 1’.

This has been corrected, thank you for your comment.

  1. Please, add information of whether data in the Table 3 is from a preclinical and/or clinical study. This will improve reading since a reader should not need to check references in the the text body to study a table.

We added a separate column clarifying this problem, thank you for your comment.

  1. Please, re-evaluate if the last row in the tables is necessary (eg. presented as ’Summary outlook’ or ’Overall Strategies’). What is the additional information as compared to the other rows presenting the detailed data?

Thank you for your comment. We agree that it is not necessary so it has been removed.

  1. It seems that in chapters 7-10 several references that already are presented in earlier chapters presenting the same topics are missing. Please, add additional references.

The references have been added, thank you.

  1. It seems that there is a writing mistake in the beginning of section 10.4, line 1-2 when writing ’… in delivering CAR cells to T cells,…’? CARs are not cells but the specific receptors engineered into T cells. Please, check the writing.

Changes have been made. Thank you for your comment.

Reviewer 2 Report

Comments and Suggestions for Authors

Review report 

The review provides a comprehensive outlook of CAR T in the context of solid tumours and the role the tumour microenvironment plays in tumour progression.

Section 1.1
The introduction to CAR T cells seems about disjointed and repetitive in some places, for example the description of the  structure of a CAR is repeated, with the first paragraph on page 3 being unnecessary. A minor point but the binding moiety of the CAR isnt always an ScFv, it can be any protein sequence that confers specificity to a target (e.g. natural ligand, VHH, peptide). There’s also mention of fourth and fifth generation CARs, as far as I’m aware the field has moved away from numeric naming after third generation as there is now such a diversity of design that it would be difficult to number and order current designs. It’s nice to see mention of other immune cell engineering like CAR-M but might also be good to mention CAR-NK as well. 

Figure 1
This is a basic cartoon of a second generation CAR, easy to understand, however the heavy and light chains of the ScFv are labelled, orientation can vary so would remove this labelling. 

Figure 2
The diagram of the tumour microenvironment is clear. I think the T-cell with the PD1-PDL1 interaction could be confusing, maybe make it clear that it’s meant to be a CTL and instead of specifying the interaction, use a flat arrow ( like a “T”) to show the inhibitory effect of the TME as this could be seen as suggesting that PD1-PDL1 is the only immunosuppressive interaction. 

Section 2.1
Section 2.1.1 provides an overview of the immune cells present in the TME, there is debate about the identity of MDSCs and TAMs and whether they’re grouped together or not but for the purpose of this review separating them is fine. The last paragraph however deals with current research into repolarising anti-inflammatory (“M2”) macrophages but there are no references, could the authors add the appropriate references? They also use the phrase “Targeting TAMs through multi-OMIC approaches” multi-OMICS isn't the right word here, I think just multiple or “a combination of” works better. 
Section 2.1.2 is very short compared to 2.1.1, they only go into detail about VEGF despite giving EGF and PDGF as other examples, might be nice to include a sentence or two about those and how they contribute to the immunosuppressive TME. Also TGFB is one of the most well studied secreted proteins in the TME and could probably do with a mention here along with other immunosuppressive cytokines. 

Section 2.1.3 describes immune checkpoints, I feel like this should more be section 2.2 as it’s the result of the immune cells and soluble factors rather than a thing on their own. 
I would like a section on the other parts of the TME that were described at the start of this section and how they contribute to immunosuppression, such as stroma and endothelial cells and ECM, there are some nice papers looking at things like the rigidity of tumours which could be included. 

Section 3
This section outlines toxicities associated with chemotherapy, radiotherapy, and CAR T. For the purpose of the review I’m not sure an entire section on this is needed as toxicities for all of these modalities have been comprehensively reviewed elsewhere. Maybe a paragraph or two on CAR T toxicity can be added to the end of the following section but I would recommend just removing this section. 

Section 4
The first 4 paragraphs of section 4 describe some of the challenges and potential methods of overcoming them for CAR T therapy in solid tumours, however the next paragraph switches to 1:1 CD4:CD8 ratios which as far as I’m aware is only implemented in one clinical product for leukaemia/lymphoma, and the reference is just a review article. 
Table 2
Honestly this table is a bit confusing, the column headers don’t really match up with the contents and I feel like it needs to be split or re-organised, e.g having one column of Findings and another of Results is confusing as they should be the same so maybe combined. Maybe also split into preclinical and clinical? 

Section 5 and 6
These sections outline CAR engineering for solid tumour efficacy, given the title I would have thought these would be the main part of the review but they’re quite short. These two sections feel like they have been written by two separate people given the same brief, they should probably be combined. A lot of the references are also review articles, it would be nice to have some primary references for each of the innovations of engineering CARs for solid tumours. A summary table like the one at the end of Section 7 might be nice as well but not completely necessary. 

Section 7
This section is good and nicely outlines the current strategies of combination therapies that include CAR T cells and the table (Table 3) is well structured and clear. 

Section 8
This section nicely outlines the different approaches for delivering CAR T and improving their trafficking to the tumour site. I feel like this would belong better as a subsection under the section of CAR engineering (Sections 5 and 6) as it incorporates engineering and is to do with the CARs themselves whereas having the combination therapies section separating these two sections doesn’t make much sense. 

Section 9
This describes clinical advances in solid tumour CAR T. For each study a description of the CAR would be nice, do they have any TME-related designs or are they all just second generation CARs? There is also a couple of paragraphs on preclinical results which I think should be moved to a different section and this one just focus on clinical results in solid tumours. If the clinical results are all just second generation it might be good to move it closer to the start of the review and use it as a nice introudction as to why we need to improve CAR T and engineer them to target or alter the TME. 

Section 10
This section isn’t really necessary, the contents should be moved to other sections, e.g. the CARs in combination with radiotherapy should be moved to the combination section and the other bits on CAR engineering should be moved to those sections; the whole review is basically a future directions so not necessary to have a section on it unless there is bleeding edge stuff that the authors feel could be the future such as a novel combination that hasnt been tried yet. 

Section 11
This provides an overall conclusion, however there are some sweeping statements, especially regarding CARs secreting IL12 and IL18 that are alluded to as the best solution to solid tumours which is doubtful as they haven’t been assessed clinically so I would remove this sentence and make broader conclusions such as promising preclinical data but yet to be proven clinically or something similar. 

Overall there is a good review in this text, it just needs a bit of reworking to make it flow and maybe a few more primary references rather than reviews. It’s not ground breaking, there are plenty of reviews of CARs in solid tumours, but it works as a broad overview for someone wanting to get into the field. 

Author Response

Dear Reviewer 2,

Thank You very much for Your time and effort, as well as constructive comments concerning our manuscript. We have studied Your comments carefully, had team discussions, sometimes difficult, and made few significant corrections and improvements, which we hope to meet with Your approval. Thanks to this work being done, we were also able to expand our knowledge even more - and enhance our future projects.

We decided to highlight all added or changed paragraphs to facilitate the process of reviewing and save Your time. Additionally, we’ve tried to explain our point of view in several points below.

We really hope these modifications can meet with Your approval.

Thank You very much for Your time and effort. We know that nowadays time is scarce. 

 Yours Sincerely,

Authors

The review provides a comprehensive outlook of CAR T in the context of solid tumours and the role the tumour microenvironment plays in tumour progression.

  1. Section 1.1 There’s also mention of fourth and fifth generation CARs, as far as I’m aware the field has moved away from numeric naming after third generation as there is now such a diversity of design that it would be difficult to number and order current designs. It’s nice to see mention of other immune cell engineering like CAR-M but might also be good to mention CAR-NK as well. 

In the last subsection of chapter 1 I did mention the CAR-M as it is the new approach to cancer treatment  and we also wrote about the research towards CAR-M in solid tumours therapy. CAR-NK has been added to this chapter. Thank you for your comment.

  1. Figure 1. This is a basic cartoon of a second generation CAR, easy to understand, however the heavy and light chains of the ScFv are labelled, orientation can vary so would remove this labelling. Add other generation as an example.

              Changes have been made and an additional figure with other generations has been added. Thank you for your comment.

  1. Figure 2. The diagram of the tumour microenvironment is clear. I think the T-cell with the PD1-PDL1 interaction could be confusing, maybe make it clear that it’s meant to be a CTL and instead of specifying the interaction, use a flat arrow ( like a “T”) to show the inhibitory effect of the TME as this could be seen as suggesting that PD1-PDL1 is the only immunosuppressive interaction. 

              Changes have been made. Thank you for your comment.

  1. Section 2.1. Section 2.1.1 provides an overview of the immune cells present in the TME, there is debate about the identity of MDSCs and TAMs and whether they’re grouped together or not but for the purpose of this review separating them is fine. The last paragraph however deals with current research into repolarising anti-inflammatory (“M2”) macrophages but there are no references, could the authors add the appropriate references? They also use the phrase “Targeting TAMs through multi-OMIC approaches” multi-OMICS isn't the right word here, I think just multiple or “a combination of” works better. 

Thank you for pointing this out. We have applied the changes and added the appropriate references as suggested.

  1. Section 2.1.2 is very short compared to 2.1.1, they only go into detail about VEGF despite giving EGF and PDGF as other examples, might be nice to include a sentence or two about those and how they contribute to the immunosuppressive TME. Also TGFB is one of the most well studied secreted proteins in the TME and could probably do with a mention here along with other immunosuppressive cytokines.

Thank you for your insightful comment. We have expanded Section 2.1.2 to include additional information on EGF, PDGF, and TGF-β 

  1. Section 2.1.3 describes immune checkpoints, I feel like this should more be section 2.2 as it’s the result of the immune cells and soluble factors rather than a thing on their own. 
    I would like a section on the other parts of the TME that were described at the start of this section and how they contribute to immunosuppression, such as stroma and endothelial cells and ECM, there are some nice papers looking at things like the rigidity of tumours which could be included. 

Thank you for your helpful comment. Section 2.2 has now been separated as suggested, and we have added additional information on stromal components, endothelial cells, and the extracellular matrix, including their roles in promoting immunosuppression and the relevance of tumour rigidity. We appreciate your valuable input.

  1. Section 3, This section outlines toxicities associated with chemotherapy, radiotherapy, and CAR T. For the purpose of the review I’m not sure an entire section on this is needed as toxicities for all of these modalities have been comprehensively reviewed elsewhere. Maybe a paragraph or two on CAR T toxicity can be added to the end of the following section but I would recommend just removing this section. 

Thank you very much for your valuable comment. After careful consideration, we have decided to retain this section in the manuscript, as we believe that including a focused discussion on CAR-T cell toxicities remains important for the context of our review. However, we have thoroughly revised the chapter and made some changes, taking into account the suggestions provided by all reviewers.

  1. Section 4. The first 4 paragraphs of section 4 describe some of the challenges and potential methods of overcoming them for CAR T therapy in solid tumours, however the next paragraph switches to 1:1 CD4:CD8 ratios which as far as I’m aware is only implemented in one clinical product for leukaemia/lymphoma, and the reference is just a review article. 

This has been included, thank you for your comment.

  1. Table 2. Honestly this table is a bit confusing, the column headers don’t really match up with the contents and I feel like it needs to be split or re-organised, e.g having one column of Findings and another of Results is confusing as they should be the same so maybe combined. Maybe also split into preclinical and clinical? 
    Thank you for your comment. We appreciate your suggestion, but after careful consideration, we have decided to retain the current format of Table 2, as we believe it allows for a comprehensive yet concise presentation of the data.
  2. Section 5 and 6. These sections outline CAR engineering for solid tumour efficacy, given the title I would have thought these would be the main part of the review but they’re quite short. These two sections feel like they have been written by two separate people given the same brief, they should probably be combined. A lot of the references are also review articles, it would be nice to have some primary references for each of the innovations of engineering CARs for solid tumours. A summary table like the one at the end of Section 7 might be nice as well but not completely necessary. 

Thank you for your thoughtful comment. We agree with your observations — Sections 5 and 6 have been revised, combined, and restructured accordingly. We also replaced several review citations with primary references and refined the coherence of the section to better align with the manuscript's focus.

  1. Section 7. This section is good and nicely outlines the current strategies of combination therapies that include CAR T cells and the table (Table 3) is well structured and clear. 

Thank you for this comment - it is especially valuable to us, as for most of us, this is our first manuscript.

  1. Section 8. This section nicely outlines the different approaches for delivering CAR T and improving their trafficking to the tumour site. I feel like this would belong better as a subsection under the section of CAR engineering (Sections 5 and 6) as it incorporates engineering and is to do with the CARs themselves whereas having the combination therapies section separating these two sections doesn’t make much sense. 

This section has been moved up to appear before the section about combination therapy. Thank you for your comment.

  1. Section 9. This describes clinical advances in solid tumour CAR T. For each study a description of the CAR would be nice, do they have any TME-related designs or are they all just second generation CARs? There is also a couple of paragraphs on preclinical results which I think should be moved to a different section and this one just focus on clinical results in solid tumours. If the clinical results are all just second generation it might be good to move it closer to the start of the review and use it as a nice introudction as to why we need to improve CAR T and engineer them to target or alter the TME. 

Thank you for your valuable comment. We added additional information for the studies. We have also revised the structure of the manuscript accordingly and relocated some of the parts.

  1. Section 10. This section isn’t really necessary, the contents should be moved to other sections, e.g. the CARs in combination with radiotherapy should be moved to the combination section and the other bits on CAR engineering should be moved to those sections; the whole review is basically a future directions so not necessary to have a section on it unless there is bleeding edge stuff that the authors feel could be the future such as a novel combination that hasnt been tried yet. 

After a team discussion, we have decided not to delete this section, as we believe it offers a concise overview of emerging directions and supports the overall structure of the manuscript. However, thank you for your comment — we have revised the manuscript structure accordingly and relocated the relevant content to the appropriate sections, while refining this part to better align with the rest of the text.

  1. Section 11. This provides an overall conclusion, however there are some sweeping statements, especially regarding CARs secreting IL12 and IL18 that are alluded to as the best solution to solid tumours which is doubtful as they haven’t been assessed clinically so I would remove this sentence and make broader conclusions such as promising preclinical data but yet to be proven clinically or something similar. 

Changes have been made. Thank you for your comment.

Overall there is a good review in this text, it just needs a bit of reworking to make it flow and maybe a few more primary references rather than reviews. It’s not ground breaking, there are plenty of reviews of CARs in solid tumours, but it works as a broad overview for someone wanting to get into the field. 

Reviewer 3 Report

Comments and Suggestions for Authors

Biology-3762586

Type of manuscript: Review
Title: Can we use CAR-T cells to overcome immunosuppression in solid tumours?
Authors: Julia Anna Gwadera, Maksymilian Grajewski, Hanna Chowaniec *, Kasper Gucia, Jagoda Michoń, Zofia Mikulicz, Malgorzata Knast, Patrycja Pujanek, Amelia Tołkacz, Aleksander Murawa, Paula Dobosz

This review paper addresses an important and timely question in cancer immunotherapy: whether CAR-T cell therapy can be effectively applied to overcome the severe immunosuppression observed in the tumor microenvironment of solid tumors. The authors aim to comprehensively and systematically present recent strategies and innovations aimed at improving CAR-T efficacy, including metabolic reprogramming, cytokine engineering, and combination therapy approaches. This review paper boasts an ambitious scope and successfully addresses key challenges and novel solutions, making it a potentially valuable resource for both academic and clinical researchers. However, additional corrections are needed for the following points. Since the paper lacks line numbers, we cannot specify where corrections should be made, so we ask that the authors thoroughly proofread the entire paper.

[Major concerns]

  1. Terminology for ‘CAR-T cells therapy’ vs. ‘CAR-T cell therapy’: "CAR-T cell" stands for "Chimeric Antigen Receptor T cell" and is treated as a singular noun. Therefore, "CAR-T cell therapy" correctly refers to a therapy that uses CAR-T cells. On the other hand, "CAR-T cells therapy" is grammatically incorrect and is not used in scientific literature.
  2. Clarify novelty in the introduction: what unique lens or framework does this review add compared to existing literature?
  3. Improve grammar and coherence throughout the manuscript through careful language editing and improved paragraph transitions.
  4. English: General compounds, proteins, etc. should be written in lowercase when used in the middle of a sentence in the text. Some keywords have their first words capitalized without needing to be capitalized, so they should all be corrected.
  5. Consider expanding the discussion on clinical translation challenges (e.g., cost, manufacturing, toxicity management).
  6. Rephrase the conclusion to provide a clearer and more polished synthesis of the main findings and future directions.
  7. Abbreviations: Abbreviations can enhance clarity and conciseness but should be used only for terms repeated frequently. Define each abbreviation by writing the full term followed by the abbreviation in parentheses at first mention, then use the abbreviation consistently throughout the paper. Define abbreviations separately in the Abstract and main text, as the Abstract is often read on its own. Use abbreviations in the Abstract only if they appear more than once. Proofread carefully to avoid redundant or inconsistent abbreviation use.
  8. In cases where abbreviations are used within figures or tables, please list these abbreviations along with their corresponding full names in the figure legends or at the bottom of corresponding tables. If there are two or more abbreviations, arrange them in alphabetical order. In this case, non-proper nouns should not have their first letters capitalized.
  9. In the list of abbreviations at the end of the paper, list the abbreviation first, followed by the full name, and list the abbreviations in alphabetical order to make it easier for readers to find the abbreviation. Organize from the perspective of the readers, not the authors.

[Minor concerns]

  1. When listing IL-18 and IL-12 together, it is appropriate to list them in Arabic numeral order, IL-12 and IL-18, unless there is a special reason.

Overall, the manuscript can be considered to publication after major revision as indicated above.

Comments on the Quality of English Language

Biology-3762586

Type of manuscript: Review
Title: Can we use CAR-T cells to overcome immunosuppression in solid tumours?
Authors: Julia Anna Gwadera, Maksymilian Grajewski, Hanna Chowaniec *, Kasper Gucia, Jagoda Michoń, Zofia Mikulicz, Malgorzata Knast, Patrycja Pujanek, Amelia Tołkacz, Aleksander Murawa, Paula Dobosz

This review paper addresses an important and timely question in cancer immunotherapy: whether CAR-T cell therapy can be effectively applied to overcome the severe immunosuppression observed in the tumor microenvironment of solid tumors. The authors aim to comprehensively and systematically present recent strategies and innovations aimed at improving CAR-T efficacy, including metabolic reprogramming, cytokine engineering, and combination therapy approaches. This review paper boasts an ambitious scope and successfully addresses key challenges and novel solutions, making it a potentially valuable resource for both academic and clinical researchers. However, additional corrections are needed for the following points. Since the paper lacks line numbers, we cannot specify where corrections should be made, so we ask that the authors thoroughly proofread the entire paper.

[Major concerns]

  1. Terminology for ‘CAR-T cells therapy’ vs. ‘CAR-T cell therapy’: "CAR-T cell" stands for "Chimeric Antigen Receptor T cell" and is treated as a singular noun. Therefore, "CAR-T cell therapy" correctly refers to a therapy that uses CAR-T cells. On the other hand, "CAR-T cells therapy" is grammatically incorrect and is not used in scientific literature.
  2. Clarify novelty in the introduction: what unique lens or framework does this review add compared to existing literature?
  3. Improve grammar and coherence throughout the manuscript through careful language editing and improved paragraph transitions.
  4. English: General compounds, proteins, etc. should be written in lowercase when used in the middle of a sentence in the text. Some keywords have their first words capitalized without needing to be capitalized, so they should all be corrected.
  5. Consider expanding the discussion on clinical translation challenges (e.g., cost, manufacturing, toxicity management).
  6. Rephrase the conclusion to provide a clearer and more polished synthesis of the main findings and future directions.
  7. Abbreviations: Abbreviations can enhance clarity and conciseness but should be used only for terms repeated frequently. Define each abbreviation by writing the full term followed by the abbreviation in parentheses at first mention, then use the abbreviation consistently throughout the paper. Define abbreviations separately in the Abstract and main text, as the Abstract is often read on its own. Use abbreviations in the Abstract only if they appear more than once. Proofread carefully to avoid redundant or inconsistent abbreviation use.
  8. In cases where abbreviations are used within figures or tables, please list these abbreviations along with their corresponding full names in the figure legends or at the bottom of corresponding tables. If there are two or more abbreviations, arrange them in alphabetical order. In this case, non-proper nouns should not have their first letters capitalized.
  9. In the list of abbreviations at the end of the paper, list the abbreviation first, followed by the full name, and list the abbreviations in alphabetical order to make it easier for readers to find the abbreviation. Organize from the perspective of the readers, not the authors.

[Minor concerns]

  1. When listing IL-18 and IL-12 together, it is appropriate to list them in Arabic numeral order, IL-12 and IL-18, unless there is a special reason.

Overall, the manuscript can be considered to publication after major revision as indicated above.

Author Response

Dear Reviewer 3,

Thank You very much for Your time and effort, as well as constructive comments concerning our manuscript. We have studied Your comments carefully, had team discussions, sometimes difficult, and made few significant corrections and improvements, which we hope to meet with Your approval. Thanks to this work being done, we were also able to expand our knowledge even more - and enhance our future projects.

We decided to highlight all added or changed paragraphs to facilitate the process of reviewing and save Your time. Additionally, we’ve tried to explain our point of view in several points below.

We really hope these modifications can meet with Your approval.

Thank You very much for Your time and effort. We know that nowadays time is scarce. 

 Yours Sincerely,

Authors

  1. Terminology for ‘CAR-T cells therapy’ vs. ‘CAR-T cell therapy’: "CAR-T cell" stands for "Chimeric Antigen Receptor T cell" and is treated as a singular noun. Therefore, "CAR-T cell therapy" correctly refers to a therapy that uses CAR-T cells. On the other hand, "CAR-T cells therapy" is grammatically incorrect and is not used in scientific literature.

The manuscript has been revised and corrected accordingly, thank you.

  1. Clarify novelty in the introduction: what unique lens or framework does this review add compared to existing literature?

Thank you for your valuable comment. Minor changes have been implemented to improve the introduction. Feedback like yours is especially important to us, as the majority of our team is at the early stage of their scientific careers, and such guidance greatly contributes to our learning and development for the future.

  1. Improve grammar and coherence throughout the manuscript through careful language editing and improved paragraph transitions.
    Thank you for your comment. We appreciate you pointing this out — the manuscript has been carefully reviewed, and grammar and paragraph transitions have been revised to improve overall coherence and clarity.
  2. English: General compounds, proteins, etc. should be written in lowercase when used in the middle of a sentence in the text. Some keywords have their first words capitalized without needing to be capitalized, so they should all be corrected.

Thank you for your observation. This issue has been reviewed and corrected throughout the manuscript to ensure consistency with standard scientific writing conventions.

  1. Consider expanding the discussion on clinical translation challenges (e.g., cost, manufacturing, toxicity management).

Thank you, we have added a new section ‘Discussion’. As this is the first manuscript for most of us, we are continuously learning, and feedback like yours helps us to improve our scientific writing skills for the future.

  1. Rephrase the conclusion to provide a clearer and more polished synthesis of the main findings and future directions.

Thank you, some changes has been made in this chapter. However, considering the suggestions from other reviewers, we have decided not to make major modifications and chose to retain the current structure of the conclusion.

  1. Abbreviations: Abbreviations can enhance clarity and conciseness but should be used only for terms repeated frequently. Define each abbreviation by writing the full term followed by the abbreviation in parentheses at first mention, then use the abbreviation consistently throughout the paper. Define abbreviations separately in the Abstract and main text, as the Abstract is often read on its own. Use abbreviations in the Abstract only if they appear more than once. Proofread carefully to avoid redundant or inconsistent abbreviation use.

Thank you for your comment and explanation of the issue. We have added full definitions of abbreviations upon first mention both in the Abstract and later -  in the main text, ensuring clarity for the readers.

  1. In cases where abbreviations are used within figures or tables, please list these abbreviations along with their corresponding full names in the figure legends or at the bottom of corresponding tables. If there are two or more abbreviations, arrange them in alphabetical order. In this case, non-proper nouns should not have their first letters capitalized.

This has been done, thank you.

  1. In the list of abbreviations at the end of the paper, list the abbreviation first, followed by the full name, and list the abbreviations in alphabetical order to make it easier for readers to find the abbreviation. Organize from the perspective of the readers, not the authors.

Thank you, we corrected the list of abbreviations by putting the abbreviation first, followed by the full name and  by placing them in alphabetical order.

  1. When listing IL-18 and IL-12 together, it is appropriate to list them in Arabic numeral order, IL-12 and IL-18, unless there is a special reason.

Thank you for pointing out this problem. This has been revised and corrected accordingly.

Round 2

Reviewer 1 Report

Comments and Suggestions for Authors

The authers have revised the most parts of the manuscript in line with my and obviosly the other reviewers’ proposals. Thank you for responding on each of my comments and questions. Your motivations and opinions are appreciated and mainly accepted by my site when evaluating this revised version. I’ll leave few major comments and proposals to improve the scientific value and to easy readers’ understanding. Also, some minor comments are included.

Major Comments

  1. The importance of the topic of tumor microenvironment (TME) in efficacy of CAR-T cell therapies have according to the authors been given a limited extend. However, the text body doesn’t leave the TME without rather comprehensive presentation of CAR-T related elements. The authors have pointed out this in the end of the subsecction 2.1. This comment is a supported solution. To focus a reader to the authors' selective presentation, the authors’ expression is proposed to be revised to a more comprehensive text including reference(s) of suitable review article of the topic TME for interested readers. Thus, please, consider my proposal for revision: ”The most important factors, already confirmed by recent research results, will be discussed here,. hHowever, the TME topic is extremely broad and also discussed by others(x,y,). requires more than one separate paper, not solely one chapter here. .
  2. Subsections 6.2 and 6.3 are to be revised by merging the most text and moving to other subsections; again to avoid the repeating text and due to length of this manuscript.
  • The 2nd section of 6.2 on local toxicity in healthy pleural tissue would suit to be presented in connection to local administration in the end of the first part of the section 6.1 to point out an example of existing limiting element as an opposite to benefits of local administration (i.e. to avoid systemic toxicity).
  • Please, revise by merging the remaining parts of subsection 6.2 and move to the subsection 6.3 as an introduction; thus, leading to deletion of the whole subsection 6.2. The Chapter 2 is a comprehensive presentation and the references 1, 120 valuable ones to an interested reader. If some given details are of importance, they can be added in a suitable contex in the subsection 6.3.

Consider a following proposal to introduce the subsection 6.3 (to be retitled as 6.2): ’The therapeutic success of CAR-T cell in solid tumours is hindered by their limited ability to traffic to and infiltrate tumour tissue due to the immunosuppressive TME expressing barriers (eg. ECM components rich tumour stroma, hypoxia, nutrient deprivation, low pH) as well as heterogeneous tumour antigen expression, antigen loss and immunosuppressive cytokines, fibrotic tumour stroma and abnormal vasculature – all expanded on in Chapter 2 (1,120).

  1. The addition of the chapter ’Discussion’ is of importance to improve the scientific value of this manuscript, and thus, has potential to create readers’ interest to specific details in the publication. Also, it should conclude and present authors’ critical evaluation of reached level of research and its implications. The following proposals are to improve the quality and understanding.

a) It’s obvious that the text is what the authors have choosen to present and evaluate. Thus, there is no need to use phrase(s) such as ’… discussed in this review…’, The introducing first sentence gives already this information. Please, revise by deleting these unneseccary wordings.

b) In the subsection 10.2, the last sentence would benefit of adding what the authers have compared when writing: ’It is worth considering that there are several similarities and differences in mechanisms, efficacy and safety profiles’. Would addition of one or the both in following ’…profiles of animal models and/or used CAR-T cell types’ be suitable according to authors point?

c) The subsection 10.2 lacks discussion on what may explain the similarities and diverse results as well as potential benefits and drawbacks of such findings.

d) The subsection 10.2 lacks discussion on important immunological differences to implement preclinical findings in humans. Furthermore, the researches need to take into account unneseccary use of animals for ethical reasons. This is also highlighted in FDA and EMA directives of development of cell and gene therapy medicinal products. The requirements of preclinical studies are limited. In contrast, there are rigorous risk-based approach based guidances how to proceed in clinical trials in human populations. The authors are strongly proposed to add this in the discussion (i.e. low value of results of in vitro bench studies and animal models in development of CAR-T cell products for therapies in humans due to artificial circumstances and immunological differences, respectively, as well as directives and guidances according to FDA and EMA, including suitable references).

e) The subsection 10.4 gives an impression of that CAR-T cells are more safe than earlier treatment modalities. This needs to be complemented with discussion of the high risk, even mortality, of immunological cytokine release syndrom and neurotoxicity requiring the therapies to be given on qualitified clinics with experience of intensive care.

Minor Comments

1. Reference(s) to CAR-NK cells is/are missing. Furthermore, the abbrerviation ’NK’ needs to be explained here and not later in subsection 2.1 (page 7/52). Please, add also in the text body whether reached efficacy in heaematological malignancies is in preclinical models or clinical trial data (chapter 1.1, page 6/52).

2. The last sentences in the introduction chapter 3 and 3.2 are obviously not necessary and thus, proposed to be deleted. The reader will easily notice of the titles of the following subsections that the topics will be discussed in details.

Please, check the whole manuscript in this aspect and delete unnecessary sentences nicely ment to introduce and/or bridge over to next section; especially due to magnitude of this review.

3. The English expression concerning the magnitude of heart complications might gain to be revised; ’big’ is mostly used to describe a subject, not an abstract fenomena. Please, consider for example: Heart-related complications reign as one of the biggest largest mortality causes in cancer survivors(68,69).

4. As already observed (my review no. 1, minor comment nr. 12) about similarity of the last rows in tables entitled as ’Summary outlook’ or ’Overall Strategies’. Please, delete the latter in table 2. The reference 117 is used in both. Thus, a reader will easily identify reached overall risks and benefits as well as remaining challenges.

5. Please, delete the sign ’:’ in the end of the title of the chapter 4 ’Engineering CAR-T Cells to Overcome Immunosuppression:

6. Chapter 8, first introducing section obviously could also refer to references 1 and 120 (not only 134) concering the tumor microenvironment? Also, concerning tumor associated antigen, references 50 and 51 would apply?

7. Chapter 8, second section on tumor antigens would benefit of adding references 50 and 51 (not only giving 114)?

8. Subsection 10.4, the first sentence writes ’CAR‑T cell therapy offers rapid treatment…’. Please, clarify /revise to make the message more clear. Do the authers mean the CAR-T cell treatment being infusion (many chemotherapies are also infusions)? The overall treatment period reaches 2-4 months at hospital that overcomes chemo and radiotherapies. Or do the authers mean CAR-T cell therapy’s early efficacy outcome, evaluable at 3-6 months?

9. Chapter 11 ’Conclusions’, the first section includes writing: ’However, a hypoxic environment can cause impairment of mitochondrial metabolism, mostly in gliomas,…’, is proposed to be revised since the knowledge is at present limited in humans due to yet few tumor microenvironment studied. Thus, for example:’… , mostly at present studied in gliomas,…

Comments on the Quality of English Language

Please, see the major comments 3a) and 3b as well as the minor comments 2, 3, 5 and 8.

Author Response

  1. The importance of the topic of tumor microenvironment (TME) in efficacy of CAR-T cell therapies have according to the authors been given a limited extend. However, the text body doesn’t leave the TME without rather comprehensive presentation of CAR-T related elements. The authors have pointed out this in the end of the subsecction 2.1. This comment is a supported solution. To focus a reader to the authors' selective presentation, the authors’ expression is proposed to be revised to a more comprehensive text including reference(s) of suitable review article of the topic TME for interested readers. Thus, please, consider my proposal for revision: ”The most important factors, already confirmed by recent research results, will be discussed here,. hHowever, the TME topic is extremely broad and also discussed by others(x,y,). requires more than one separate paper, not solely one chapter here.

Thank you for your valuable comment. We agree that the tumour microenvironment is a broad topic and that referencing comprehensive reviews would improve clarity. We have revised the text accordingly and added references to guide readers to further resources. We believe these changes enhance the manuscript and appreciate your suggestion.

  1. Subsections 6.2 and 6.3 are to be revised by merging the most text and moving to other subsections; again to avoid the repeating text and due to length of this manuscript.
  • The 2nd section of 6.2 on local toxicity in healthy pleural tissue would suit to be presented in connection to local administration in the end of the first part of the section 6.1 to point out an example of existing limiting element as an opposite to benefits of local administration (i.e. to avoid systemic toxicity).
  • Please, revise by merging the remaining parts of subsection 6.2 and move to the subsection 6.3 as an introduction; thus, leading to deletion of the whole subsection 6.2. The Chapter 2 is a comprehensive presentation and the references 1, 120 valuable ones to an interested reader. If some given details are of importance, they can be added in a suitable contex in the subsection 6.3.

Consider a following proposal to introduce the subsection 6.3 (to be retitled as 6.2): ’The therapeutic success of CAR-T cell in solid tumours is hindered by their limited ability to traffic to and infiltrate tumour tissue due to the immunosuppressive TME expressing barriers (eg. ECM components rich tumour stroma, hypoxia, nutrient deprivation, low pH) as well as heterogeneous tumour antigen expression, antigen loss and immunosuppressive cytokines, fibrotic tumour stroma and abnormal vasculature – all expanded on in Chapter 2 (1,120).

Thank you for your comment and the proposal of modification, we have modified the text accordingly.

  1. The addition of the chapter ’Discussion’ is of importance to improve the scientific value of this manuscript, and thus, has potential to create readers’ interest to specific details in the publication. Also, it should conclude and present authors’ critical evaluation of reached level of research and its implications. The following proposals are to improve the quality and understanding.
  1. a) It’s obvious that the text is what the authors have choosen to present and evaluate. Thus, there is no need to use phrase(s) such as ’… discussed in this review…’, The introducing first sentence gives already this information. Please, revise by deleting these unneseccary wordings.

Thank you for your comment, we agree that this proposal will improve the quality and understanding so this has been done.

  1. b) In the subsection 10.2, the last sentence would benefit of adding what the authers have compared when writing: ’It is worth considering that there are several similarities and differences in mechanisms, efficacy and safety profiles’. Would addition of one or the both in following ’…profiles of animal models and/or used CAR-T cell types’ be suitable according to authors point?

We think that this addition would be suitable, so we made this addition in the manuscript. Thank you for this suggestion.

  1. c) The subsection 10.2 lacks discussion on what may explain the similarities and diverse results as well as potential benefits and drawbacks of such findings.

Thank you for your comment, we agree that including the similarities and diverse resulte will come with a benefit to the Discussion section; this has been done.

  1. d) The subsection 10.2 lacks discussion on important immunological differences to implement preclinical findings in humans. Furthermore, the researches need to take into account unneseccary use of animals for ethical reasons. This is also highlighted in FDA and EMA directives of development of cell and gene therapy medicinal products. The requirements of preclinical studies are limited. In contrast, there are rigorous risk-based approach based guidances how to proceed in clinical trials in human populations. The authors are strongly proposed to add this in the discussion (i.e. low value of results of in vitro bench studies and animal models in development of CAR-T cell products for therapies in humans due to artificial circumstances and immunological differences, respectively, as well as directives and guidances according to FDA and EMA, including suitable references).

We discussed your proposition and agreed on adding this to the Discussion chapter. Thank you for your comment.

  1. e) The subsection 10.4 gives an impression of that CAR-T cells are more safe than earlier treatment modalities. This needs to be complemented with discussion of the high risk, even mortality, of immunological cytokine release syndrom and neurotoxicity requiring the therapies to be given on qualitified clinics with experience of intensive care.

Thank you for your comment. We have expanded the discussion on this topic accordingly.

Minor Comments

  1. Reference(s) to CAR-NK cells is/are missing. Furthermore, the abbrerviation ’NK’ needs to be explained here and not later in subsection 2.1 (page 7/52). Please, add also in the text body whether reached efficacy in heaematological malignancies is in preclinical models or clinical trial data (chapter 1.1, page 6/52).

We added the references, as well as explained the abbreviation. We also agree that the section would benefit from adding information whether it is a clinical trial or preclinical model and we did that. Thank you for your comment and corrections. 

  1. The last sentences in the introduction chapter 3 and 3.2 are obviously not necessary and thus, proposed to be deleted. The reader will easily notice of the titles of the following subsections that the topics will be discussed in details.

We want to make sure that our manuscript is accessible and pleasant to read, so we made changes. Thank you for your thoughtful comment. This type of guidance is important to us, as we are still learning  to write scientific papers, so thank you very much.

Please, check the whole manuscript in this aspect and delete unnecessary sentences nicely ment to introduce and/or bridge over to next section; especially due to magnitude of this review.

Thank you for your comment, we support your opinion that a reader will notice the related chapter,  this has been done.

  1. The English expression concerning the magnitude of heart complications might gain to be revised; ’big’ is mostly used to describe a subject, not an abstract fenomena. Please, consider for example: Heart-related complications reign as one of the biggest largest mortality causes in cancer survivors(68,69).

Thank you, the correction has been made. We are aware, that some language mistakes can be present, as English is not our native language, though we do our best to correct all the mistakes so thank you for pointing this one out

  1. As already observed (my review no. 1, minor comment nr. 12) about similarity of the last rows in tables entitled as ’Summary outlook’ or ’Overall Strate gies’. Please, delete the latter in table 2. The reference 117 is used in both. Thus, a reader will easily identify reached overall risks and benefits as well as remaining challenges.

The section has been removed; thank you for your valuable comment. We fully agree that this revision enhances clarity and improves the overall readability of the table.

  1. Please, delete the sign ’:’ in the end of the title of the chapter 4 ’Engineering CAR-T Cells to Overcome Immunosuppression:

This has been done, thank you.

  1. Chapter 8, first introducing section obviously could also refer to references 1 and 120 (not only 134) concering the tumor microenvironment? Also, concerning tumor associated antigen, references 50 and 51 would apply?
  2. Chapter 8, second section on tumor antigens would benefit of adding references 50 and 51 (not only giving 114)?

Thank you, we agree that adding reference no. 50 is a good idea and we did so.

  1. Subsection 10.4, the first sentence writes ’CAR‑T cell therapy offers rapid treatment…’. Please, clarify /revise to make the message more clear. Do the authers mean the CAR-T cell treatment being infusion (many chemotherapies are also infusions)? The overall treatment period reaches 2-4 months at hospital that overcomes chemo and radiotherapies. Or do the authers mean CAR-T cell therapy’s early efficacy outcome, evaluable at 3-6 months?

Thank you for your comment, we have revised the subsection and clarified the information.

  1. Chapter 11 ’Conclusions’, the first section includes writing: ’However, a hypoxic environment can cause impairment of mitochondrial metabolism, mostly in gliomas,…’, is proposed to be revised since the knowledge is at present limited in humans due to yet few tumor microenvironment studied. Thus, for example:’… , mostly at present studied in gliomas,…

Thank you, we revised this part the correction has been made

Reviewer 3 Report

Comments and Suggestions for Authors

biology-3762586-v2

Type of manuscript: Review
Title: Can we use CAR-T cells to overcome immunosuppression in solid tumours?
Authors: Julia Anna Gwadera, Maksymilian Grajewski, Hanna Chowaniec *, Kasper Gucia, Jagoda Michoń, Zofia Mikulicz, Malgorzata Knast, Patrycja Pujanek, Amelia Tołkacz, Aleksander Murawa, Paula Dobosz

Most of the points pointed out during the first review process have been appropriately reflected and corrected, and the level of the paper has also greatly improved. In addition, the points pointed out can be corrected during the proofreading process, so please cooperate with the office to correct them well.

One more recommendation to the authors is that papers for submission should have page numbers and line numbers as a basic requirement.

[Major concerns]

  1. English: This is about the notation of 'tumor' and 'tumour'. As you know, 'tumor' is American English and 'tumour' is British English. Including the references, 'tumor' was used a total of 20 times, and of those, it was used twice in the text, while 'tumour' was used a total of 280 times including the references. Even if the references are left as is, it is correct to choose one of the two notations in the text. Therefore, in this paper, it seems that 'tumor' used twice in the text should be changed to 'tumour'.
  2. Tables 1~3: One of the principles of writing scientific papers is that the title of a table is placed at the top of the table, and the title of a figure is placed at the bottom of the figure. When making a table, if both the horizontal and vertical lines are solid lines, it will be too stuffy and difficult for readers to read the table. Therefore, horizontal lines must be present, but vertical lines should be deleted if possible.
  3. I hope that other minor typos will be corrected with Office during the proofreading process.

Overall, the manuscript can be considered to publication after minor revision as indicated above.

Comments on the Quality of English Language

biology-3762586-v2

Type of manuscript: Review
Title: Can we use CAR-T cells to overcome immunosuppression in solid tumours?
Authors: Julia Anna Gwadera, Maksymilian Grajewski, Hanna Chowaniec *, Kasper Gucia, Jagoda Michoń, Zofia Mikulicz, Malgorzata Knast, Patrycja Pujanek, Amelia Tołkacz, Aleksander Murawa, Paula Dobosz

Most of the points pointed out during the first review process have been appropriately reflected and corrected, and the level of the paper has also greatly improved. In addition, the points pointed out can be corrected during the proofreading process, so please cooperate with the office to correct them well.

One more recommendation to the authors is that papers for submission should have page numbers and line numbers as a basic requirement.

[Major concerns]

  1. English: This is about the notation of 'tumor' and 'tumour'. As you know, 'tumor' is American English and 'tumour' is British English. Including the references, 'tumor' was used a total of 20 times, and of those, it was used twice in the text, while 'tumour' was used a total of 280 times including the references. Even if the references are left as is, it is correct to choose one of the two notations in the text. Therefore, in this paper, it seems that 'tumor' used twice in the text should be changed to 'tumour'.
  2. Tables 1~3: One of the principles of writing scientific papers is that the title of a table is placed at the top of the table, and the title of a figure is placed at the bottom of the figure. When making a table, if both the horizontal and vertical lines are solid lines, it will be too stuffy and difficult for readers to read the table. Therefore, horizontal lines must be present, but vertical lines should be deleted if possible.
  3. I hope that other minor typos will be corrected with Office during the proofreading process.

Overall, the manuscript can be considered to publication after minor revision as indicated above.

Author Response

  1. English: This is about the notation of 'tumor' and 'tumour'. As you know, 'tumor' is American English and 'tumour' is British English. Including the references, 'tumor' was used a total of 20 times, and of those, it was used twice in the text, while 'tumour' was used a total of 280 times including the references. Even if the references are left as is, it is correct to choose one of the two notations in the text. Therefore, in this paper, it seems that 'tumor' used twice in the text should be changed to 'tumour'.

We fully agree, we will make sure to correct the mistakes and use British English only. Thank you.

  1. Tables 1~3: One of the principles of writing scientific papers is that the title of a table is placed at the top of the table, and the title of a figure is placed at the bottom of the figure. When making a table, if both the horizontal and vertical lines are solid lines, it will be too stuffy and difficult for readers to read the table. Therefore, horizontal lines must be present, but vertical lines should be deleted if possible.

Thank you for your comment, as we are still learning about writing scientific papers such notes help us in learning and becoming more accurate in our work.

  1. I hope that other minor typos will be corrected with Office during the proofreading process.

We will also make sure to proofread and correct the typos, thank you for your comment.